# Towards Scalable and Consistent 3D Editing

**Ruihao Xia** [1 2] **Yang Tang** [1] **Pan Zhou** [2]

## Abstract

3D editing—the task of locally modifying the geometry or appearance of a 3D asset—has wide applications in immersive content creation, digital entertainment, and AR/VR. However, unlike 2D editing, it remains challenging due to the need for cross-view consistency, structural fidelity, and fine-grained controllability. Existing approaches are often slow, prone to geometric distortions, or dependent on manual and accurate 3D masks that are error-prone and impractical. To address these challenges, we advance both the data and model fronts. On the data side, we introduce **3DEdit-Verse**, the largest paired 3D editing benchmark to date, comprising 116,309 high-quality training pairs and 1,500 curated test pairs. Built through complementary pipelines of pose-driven geometric edits and foundation model-guided appearance edits, 3DEditVerse ensures edit locality, multi-view consistency, and semantic alignment. On the model side, we propose **3DEditFormer**, a 3D-structure-preserving transformer. By enhancing image-to-3D generation with dual-guidance attention and time-adaptive gating, 3DEditFormer disentangles editable regions from preserved structure, enabling precise and consistent edits without requiring auxiliary 3D masks. Extensive experiments demonstrate that our framework outperforms state-of-the-art baselines both quantitatively and qualitatively, establishing a new standard for practical and scalable 3D editing. Dataset and code are available at https://www.lv-lab.org/3DEditFormer/

## 1. Introduction

3D editing manipulates geometry and appearance locally, enabling critical applications in immersive content creation (Chen et al., 2018), digital entertainment (Zhan et al., 2024), AR/VR (Madhavaram et al., 2025), and product design (Huang et al., 2024a). Despite its importance, 3D editing remains far more challenging than 2D editing (Brooks et al., 2023; Kawar et al., 2023; Liu et al., 2025; Wang et al., 2025; Huang et al., 2025a). Unlike 2D editing, which modifies a single image, 3D editing must simultaneously ensure cross-view geometric consistency, global structural fidelity, and fine-grained controllability. These challenges have prevented 3D editing from reaching the ease and accessibility of modern 2D tools.

Existing approaches follow three main paradigms. First, *Score Distillation Sampling (SDS)* (Dong et al., 2024; Liu et al., 2024; Zhuang et al., 2024) distills 2D diffusion priors (Rombach et al., 2022) but is prohibitively slow. Second, *multi-view image editing methods* (Chen et al., 2024a; Bar-On et al., 2025; Erkoç et al., 2025) modify rendered views and reconstruct them into 3D (Xu et al., 2024a), yet struggle with distortions and cross-view misalignments. Third, *end-to-end generative models* (Xiang et al., 2025; Li et al., 2025a) operate in latent spaces but rely on coarse, manual 3D masks that often cause unintended global alterations. For example, in Fig. 4 (4th column), adding a hat to a dog also alters its body. These limitations make existing methods impractical for real-world use.

Ideally, a practical 3D editing system should match the intuitiveness of modern 2D editing tools (Liu et al., 2025; Wang et al., 2025): allowing users to specify edits with simple prompts while producing fast, precise, localized, and structure-preserving modifications, without manual mask creation. We follow the established *image-guided 3D editing* paradigm (Bar-On et al., 2025; Li et al., 2025a), where the user first modifies a reference view using any 2D editor. The key challenge, then, is: *How can we enable precise, localized 3D edits with intuitive prompts while maintaining structural fidelity across views?* This is the problem we tackle in this work.

**Contributions.** To address this challenge, we focus on two fundamental bottlenecks: the scarcity of paired 3D editing datasets and the difficulty of achieving controllable

---

[1]Key Laboratory of Smart Manufacturing in Energy Chemical Process, Ministry of Education, East China University of Science and Technology, Shanghai, China [2]School of Computing and Information Systems, Singapore Management University, Singapore. Correspondence to: Yang Tang <yangtang@ecust.edu.cn>, Pan Zhou <panzhou@smu.edu.sg>.

*Proceedings of the 43rd International Conference on Machine Learning*, Seoul, South Korea. PMLR 306, 2026. Copyright 2026 by the author(s).

*Table 1.* 3D editing dataset comparison of 3D-Alpaca-Editing (Ye et al., 2025), CMD (Li et al., 2025b), and Edit3D-Bench (Li et al., 2025a) across key criteria. "Consistency" refers to the preservation of unedited regions. 3D-Alpaca-Editing lacks this property as it independently generates the before/after assets. "Harmony" denotes whether the edit appears natural and semantically coherent. CMD falls short in this regard since it constructs new objects by concatenating unrelated 3D assets.

| 3D Editing Datasets | Training Size | Testing Size | Edit Region | Training & Test | Consistency | Harmony |
|---|---|---|---|---|---|---|
| 3D-Alpaca-Editing | 52,532 | — | ✗ | ✗ | ✗ | ✔ |
| CMD | 40,000 | 50 | ✗ | ✔ | ✔ | ✗ |
| Edit3D-Bench | — | 300 | ✔ | ✗ | ✔ | ✔ |
| 3DEditVerse (Ours) | 116,309 | 1,500 | ✔ | ✔ | ✔ | ✔ |

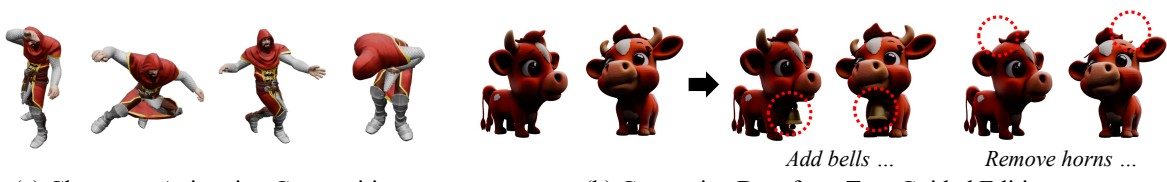

(a) Character–Animation Compositions         (b) Generative Data from Text-Guided Editing

*Figure 1.* Some examples of our 3DEditVerse dataset. See more examples in Appendix A.

and structure-preserving edits. Our main contributions are highlighted below.

First, we introduce **3DEditVerse**, the first large-scale high-fidelity 3D editing benchmark, comprising 116,309 paired original and edited 3D assets for training and 1,500 for testing. Unlike prior datasets (see Tab. 1) which are limited in scale, edit diversity, or annotation granularity, 3DEditVerse meets four essential criteria: localized edit regions, scalability for large-scale training, multi-view consistency, and semantic harmony. It is constructed through two complementary pipelines: (i) pose-driven geometric edits, which generate "before–after" assets capturing diverse articulations and geometric variations of animated characters; (ii) appearance-driven edits, guided by textual instructions and leveraging a cascade of foundation models—DeepSeek-R1 (Guo et al., 2025) for prompt diversification, Flux (Labs, 2024; Labs et al., 2025) for source-target image synthesis, Qwen-VL (Bai et al., 2025) for automated edit instruction generation and region localization, and Trellis (Xiang et al., 2025) for 3D lifting—augmented with multi-view mask projection and latent-space repainting (Lugmayr et al., 2022). Finally, 1,500 test samples are manually evaluated and carefully curated through human assessment. This design ensures edits are localized, consistent across views, and harmonious with unedited regions, providing the first scalable high-quality resource for training and evaluating end-to-end 3D editing models.

Second, we propose the **3D-structure-preserving transformer (3DEditFormer)**, a novel extension of image-to-3D Trellis (Xiang et al., 2025) tailored for 3D editing. Existing image-to-3D diffusion models (Xiang et al., 2025; Yang et al., 2024) can generate plausible assets but struggle to

preserve structure: unedited regions often drift, and source-target image guidance alone is insufficient to maintain geometric and textural fidelity. 3DEditFormer addresses this by injecting multi-stage features from the source asset into target generation. Specifically, we design a Dual-Guidance Attention Block with two cross-attention pathways: one attends to fine-grained structural features at late diffusion steps, while the other attends to semantic transition features at early steps. A Time-Adaptive Gating mechanism balances their influence to emphasize semantic edits early and structural fidelity later. This enables localized, consistent, and structure-preserving 3D edits without manual 3D masks (Barda et al., 2025) or external constraints (Li et al., 2025a).

Finally, training 3DEditFormer on 3DEditVerse achieves state-of-the-art (SoTA) 3D editing performance. Our approach produces edits that are both faithful to user intent and consistent across views. As shown in Fig. 4, our approach enables high-quality local modifications while maintaining structural fidelity, outperforming existing baselines by significant margins. Moreover, unlike VoxHammer (Li et al., 2025a), which relies on precise 3D masks as auxiliary input, our 3DEditFormer achieves superior results without requiring any mask, yielding an average +13% improvement on 3D metrics and demonstrating both higher fidelity and greater practicality.

## 2. Related Work

**3D Generation.** Early 3D generation relied on GANs (Goodfellow et al., 2020) but struggled with diversity and fidelity (Chan et al., 2022; Gao et al., 2022). Diffusion models (Ho et al., 2020) later improved quality across dif-

ferent representations such as multi-view images (Liu et al., 2023; Huang et al., 2024c), triplanes (Wu et al., 2024a; Shue et al., 2023), and 3D Gaussians (Chen et al., 2024c; Xu et al., 2024b), yet efficiency and accurate appearance modeling remain challenging issues. Recent works (Xiang et al., 2025; Yang et al., 2024) move toward native or latent 3D spaces, e.g., 3DTopia-XL (Chen et al., 2025) leverages a primitive-based PBR-ready representation and a DiT-based pipeline for high-quality 3D asset synthesis, GaussianAnything (Yushi et al., 2025) adopts a point-cloud-structured latent space with cascaded latent flows for multi-modal 3D Gaussian generation, and 3DShape2VecSet (Zhang et al., 2023) encodes shapes as neural fields over a set of latent vectors tailored to transformers for versatile 3D generative modeling. We build on these advances and extend latent-space diffusion to localized editing, introducing structural priors and edit-aware mechanisms that enable faithful, structure-preserving modifications.

**3D Editing.** Existing approaches fall into three main paradigms. SDS-based methods (Miao et al., 2025; Huang et al., 2025b) leverage 2D diffusion priors to optimize 3D assets, but are prohibitively slow and unsuitable for interactive use. Multi-view editing methods (Chen et al., 2024b; Qi et al., 2024) modify rendered images before reconstructing them into 3D, offering efficiency but often suffering from cross-view inconsistencies and distorted geometry. End-to-end generative models directly edit assets in latent space, achieving better integration of shape and texture, but they still depend on manually annotated 3D masks (Xiang et al., 2025; Li et al., 2025a), which are coarse, labor-intensive, and prone to unintended modifications. In contrast, our 3DEditFormer eliminates the need for such manual masks while still achieving localized and consistent 3D edits.

## 3. 3DEditVerse Dataset

A critical obstacle in advancing 3D editing is the absence of large-scale paired datasets of original and edited 3D assets. Such pairs are essential for training models that can faithfully learn how local edits affect geometry and appearance while preserving unedited regions. Without them, models either overfit to synthetic toy edits or rely on weak supervision, limiting generalization and practical use. Existing datasets (Li et al., 2025b;a; Ye et al., 2025), summarized in Tab. 1, are insufficient due to small scale, missing edit correspondences, or unrealistic scenarios. This gap prevents 3D editing methods from achieving the same accessibility and precision as their 2D counterparts.

To address this, we present **3DEditVerse**, the first large-scale dataset of paired 3D assets for local editing. It is designed to (i) provide sufficient scale and diversity, covering both geometric and appearance edits, and (ii) ensure edits are realistic, localized, and structure-preserving. To this end,

we introduce two automated pipelines—pose-driven geometric edits and text-guided appearance edits—that generate high-quality paired assets at scale. In addition, 1,500 test samples are manually evaluated and curated to guarantee the reliability of evaluation benchmark.

### 3.1. Character–Animation Compositions for Geometric Edits

The first pipeline targets pose-driven *geometric edits*, where the same object undergoes articulation or structural variation while maintaining identity. We leverage publicly available 3D characters and animation sequences (Inc., 2021), exploiting the fact that different poses of the same character naturally form valid "before–after" edit pairs. The generation process proceeds in two steps.

**(1) Candidate Pose Generation.** Animation sequences are sampled at fixed intervals to extract candidate frames. A key challenge is redundancy: many poses across or within sequences are visually similar. To ensure diversity, we render each pose from a canonical view, extract embeddings using DINOv2 (Oquab et al., 2024), and prune near-duplicates based on cosine similarity. This yields a curated pool of 4,998 unique candidate poses spanning a wide variety of articulations.

**(2) Data Assembly.** From our collection of 108 distinct characters, we pair each one with 500 poses randomly selected from the candidate pool. This procedure results in a total of $108 \times 500 = 54,000$ unique 3D assets. Paired data is then formed by associating different pose-renders of the same character, providing a rich source for training models on pose and shape alterations. The visualization of character–animation compositions is shown in Fig. 1.

### 3.2. Text-Guided Generation for Appearance Edits

The second pipeline focuses on *appearance-driven edits*, which modify textures, colors, or fine details while preserving overall geometry. Unlike geometry edits, these require sophisticated generative pipelines. We design a fully automated text-to-image-to-3D lifting pipeline via a cascade of foundation models. In Fig. 2, each model is annotated with a numbered label (e.g., ①), and we follow the same numbering in the description for clarity. Details of the instructional prompts for each model are provided in Appendix B.

**Source and Target Images, and Edit Prompt Generation.** As shown in Fig. 2 (upper), we start with the 4,585-word vocabulary from (Huang et al., 2024b). For each word, ① DeepSeek-R1 (Guo et al., 2025) generates diverse descriptive prompts, which are fed into ② Flux.1-Dev (Labs, 2024) to synthesize a high-quality source image $I^{src}$. This serves as the "before-edit" state. Then, ③ Qwen-VL (Bai et al., 2025) analyzes $I^{src}$ and generates edit instructions $p^{edit}$, each

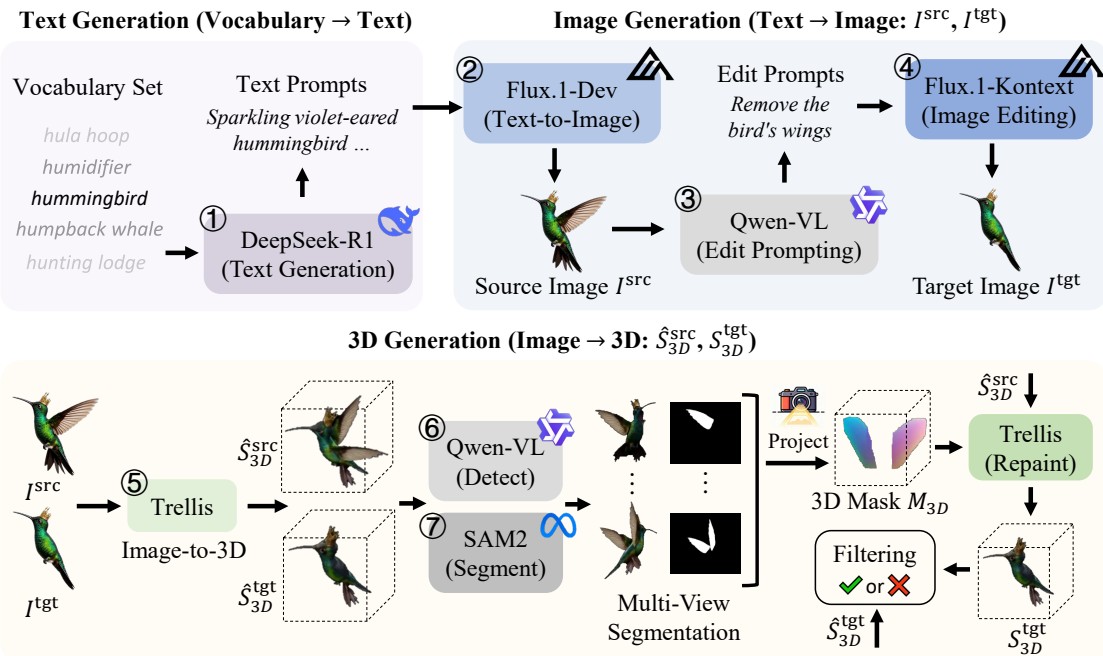

*Figure 2.* Overview of our data generation pipeline for text-guided 3D editing. Starting from a large-scale Vocabulary Set, we employ multiple foundation models in a carefully orchestrated manner and construct the text-to-image-to-3D lifting pipeline.

describing a plausible and semantically coherent modification (e.g., "add a vase to the table"). Finally, $I^{\text{src}}, p^{\text{edit}}$ is provided to ④ Flux.1-Kontext (Labs et al., 2025), which executes the edit and outputs the target image $I^{\text{tgt}}$. This produces large-scale, semantically rich image editing pairs for lifting into 3D.

In addition, we leverage samples from (Ye et al., 2025), where each sample provides a source image rendered from existing 3D assets in the Objaverse-XL dataset (Deitke et al., 2023) and an accompanying edit prompt generated from a predefined template. We take these source–prompt pairs and apply the Flux.1-Kontext model (Labs et al., 2025) to execute the edits. This augmentation not only diversifies the distribution of editing instructions but also ensures alignment with real 3D geometries, strengthening the robustness of our paired data.

**3D Lifting with Consistency Preservation.** A straightforward approach of independently lifting the source and target images to 3D using models such as Trellis (Xiang et al., 2025) often produces severe geometric distortions and identity mismatches. To address this, we propose a consistency-preserving 3D lifting pipeline that explicitly localizes the edit region in 3D and applies a mask-guided repainting strategy (Lugmayr et al., 2022) to ensure fidelity.

**(1)** Edited-Region Identification. As shown in Fig. 2 (lower-left), we first generate initial 3D assets $\hat{S}_{3D}^{\text{src}}$ and $\hat{S}_{3D}^{\text{tgt}}$ from the source and target images with the Trellis model. To

localize the edit without manual annotation, we employ ⑤ Qwen-VL as an open-set detector: given a rendered view of the 3D asset and the edit instruction, it outputs a 2D bounding box $B_{2D}$ to highlight the edited region.

(2) Multi-View 3D Mask Projection. As shown in Fig. 2 (lower), to obtain a robust 3D mask, we render the rotating asset $\hat{S}_{3D}^{src/tgt}$ into a sequence of views and apply ⑥ SAM2 (Ravi et al., 2025) to segment and track the target region across frames. The resulting 2D masks $\{M_{2D}^i\}$ are then back-projected into 3D space using the pinhole camera model. Formally, given the intrinsic and extrinsic parameters $K_i$ and $[R_i \mid t_i]$ of camera $i$, one can project a voxel $v = (x, y, z, 1)^\top$ onto the $i$-th view:

$$\tilde{p}_i = K_i[R_i \mid t_i]v, \quad p_i = \left(\frac{\tilde{p}_{i,x}}{\tilde{p}_{i,z}}, \frac{\tilde{p}_{i,y}}{\tilde{p}_{i,z}}\right). \quad (1)$$

We check whether $p_i$ lies inside the 2D mask $M_{2D}^i$, and accumulate evidence across all views:

$$c(v) = \sum_{i=1}^{N} \mathbb{1}[p_i \in M_{2D}^i], \quad (2)$$

where $N$ is the number of views, and is set to 70 in this work. The final 3D mask is defined as

$$M_{3D} = \{v \mid c(v) \geq \tau\}, \quad (3)$$

retaining voxels consistently supported by at least a fraction $\tau$ of views. This ensures that the mask is geometrically consistent and resilient to segmentation noise.

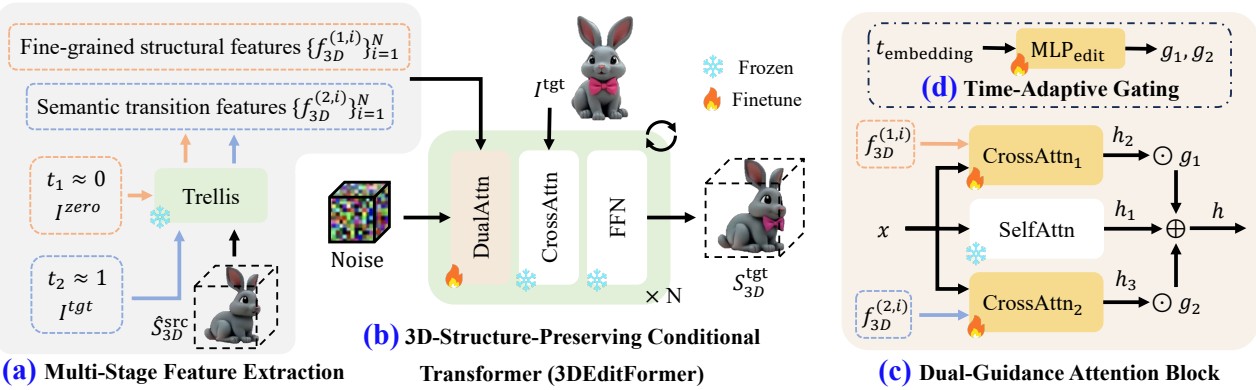

*Figure 3.* Overview of our proposed 3DEditFormer. (a) Multi-stage features $\{f_{3D}^{(1,i)}\}_{i=1}^N$ and $\{f_{3D}^{(2,i)}\}_{i=1}^N$ are extracted from the frozen Trellis model (Xiang et al., 2025) at different denoising timesteps, capturing fine-grained structural priors and semantic transition cues, respectively. (b) These features are injected into each transformer layer via (c) Dual-Guidance Attention Block, where their contributions are modulated by (d) Time-Adaptive Gating mechanism.

**(3)** Localized 3D Editing. With $M_{3D}$, we perform localized 3D editing using the Repaint strategy (Lugmayr et al., 2022) within Trellis (Xiang et al., 2025), as shown in Fig. 2 (lower-right). Specifically, during the denoising process of generating the target 3D asset $S_{3D}^{\text{tgt}}$, we inject controlled noise into the source asset $\hat{S}_{3D}^{\text{src}}$ to obtain a noisy latent:

$$z_t^{\text{src}} \sim \mathcal{N}\big(\hat{S}_{3D}^{\text{src}}, \sigma_t\big), \qquad (4)$$

where $\sigma_t$ denotes the noise variance at timestep $t$. To localize editing, we fuse the source and target latents using the binary 3D mask $M_{3D}$:

$$\hat{z}_t = M_{3D} \odot z_t^{\text{tgt}} + (1 - M_{3D}) \odot z_t^{\text{src}}, \qquad (5)$$

where $\odot$ denotes element-wise multiplication. In this process, voxels inside the editing region are updated according to the evolving target latent $z_t^{\text{tgt}}$, while voxels outside remain anchored to the source latent $z_t^{\text{src}}$. This mechanism ensures precise modifications strictly within the masked region while preserving the rest of the geometry, yielding high-fidelity structure-preserving edits.

**(4)** Post-Editing Consistency Filtering. To ensure global consistency, in Fig. 2 (lower-right), we render both the edited asset $S_{3D}^{\text{tgt}}$ and the initial target prediction $\hat{S}_{3D}^{\text{tgt}}$ into multiple views and compare them using DINOv2 (Oquab et al., 2024) feature similarity. Samples with mean cosine similarity below a threshold are discarded, effectively removing artifacts from incomplete mask coverage and enhancing the robustness of the pipeline.

Together, the two pipelines produce about 118K paired 3D assets, covering both 54,000 structural pose-driven and 64,123 appearance-based edits. Crucially, all pairs are localized, consistent across views, and semantically coherent, enabling robust supervised training of 3D editing models.

Then, 1,500 test samples are manually assessed and carefully curated to further guarantee the reliability of the evaluation benchmark. As shown in Tab. 1, unlike existing datasets that are either too small or weakly annotated, 3DEditVerse is the first benchmark to combine scale, diversity, and fidelity. This resource lays the foundation for systematic progress in 3D editing research.

## 4. 3D-Structure-Preserving Transformer

While SoTA image-to-3D generation models (Xiang et al., 2025; Yang et al., 2024; Wu et al., 2024c) can synthesize plausible 3D assets from a single image, they struggle to preserve structural consistency in editing scenarios. In particular, providing only the source or target images is insufficient for the model to determine which regions of the original geometry and texture should remain unchanged, often leading to unintended distortions in unedited areas.

To address this problem, we introduce the **3D-Structure-Preserving Transformer (3DEditFormer)**, a new framework explicitly designed to inject structural priors from the source asset into the generation of the edited asset. Unlike prior approaches that treat editing as re-synthesis from scratch, 3DEditFormer enforces a principled coupling between source and target through dual structural guidance. As shown in Fig. 3, our framework consists of three key innovations: (i) a **Dual-Guidance Attention Block** that integrates source-aware cross-attention at multiple levels, (ii) a **Multi-Stage Feature Extraction** module that disentangles fine-grained structural fidelity from semantic transition cues, and (iii) a **Time-Adaptive Gating** mechanism that dynamically balances these signals across denoising stages. Together, these components resolve the inconsistency problem of prior methods and enable edits that are both localized and structure-preserving in 3D space.

## 4.1. Dual-Guidance Attention Block

3DEditFormer builds upon the Trellis (Xiang et al., 2025), an image-to-3D framework which stacks $N$ transformer attention layers consisting of self-attention, cross-attention, and Feed-Forward Networks (FFN). We freeze the Trellis backbone to retain its generative strength and augment the original self-attention with our proposed Dual-Guidance Attention Block (DualAttn), as shown in Fig. 3 (b). This block introduces two parallel cross-attention branches, while keeping the other pathways untouched. These two cross-attention branches interact with the multi-stage features described in Sec. 4.2, which encode complementary structural information from the source 3D asset, as shown in Fig. 3 (c). Accordingly, 3DEditFormer directly injects source-aware priors into every layer, constraining the editing process to remain faithful to the original structure of the source 3D asset.

**Distinctive Novelty from Traditional Dual Attention**. It is crucial to highlight that our *Dual-Guidance* Attention mechanism is fundamentally distinct from conventional "dual attention" frameworks (e.g., spatial-channel dual attention architectures (Fu et al., 2019; Ding et al., 2022)) in both motivation and execution. While traditional methods focus on capturing static multi-dimensional dependencies (such as spatial vs. channel layouts) within a single feature map, our block is uniquely designed around the *temporal dynamics* of the diffusion process. Instead of operating on spatial dimensions, it introduces parallel cross-attention branches to decouple fine-grained structural features from semantic transition cues, resolving the core challenge of geometry-consistent 3D asset editing.

Formally, let $x$ be the input feature of the $i$-th dual-guidance attention block. Each block first computes:

$$h_1 = \text{SelfAttn}(\text{Norm}(x)), \tag{6}$$

representing the original frozen self-attention pathway. To integrate source 3D structural priors, we introduce two additional feature sets, $\{f_{3D}^{(1,i)}, f_{3D}^{(2,i)}\}$, extracted from the frozen Trellis at distinct timesteps (see details in Sec. 4.2). Then, the corresponding cross-attention branches are:

$$
\begin{aligned}
h_2 &= \text{CrossAttn}_1(\text{Norm}(x), f_{3D}^{(1,i)}), \\
h_3 &= \text{CrossAttn}_2(\text{Norm}(x), f_{3D}^{(2,i)}).
\end{aligned}
\tag{7}
$$

The outputs are adaptively gated using timestep-dependent coefficients $(g_1, g_2)$, which will be elaborated on Sec. 4.3:

$$h = h_1 + g_1 \odot h_2 + g_2 \odot h_3, \tag{8}$$

where $\odot$ denotes element-wise scaling. This fused representation $h$ is then passed through the original cross-attention with image context $I^{\text{tgt}}$ and the FFN, completing the attention layer computation.

## 4.2. Multi-Stage Feature Extraction

A central novelty of 3DEditFormer lies in its dual feature design, which captures complementary signals from different diffusion stages, as shown in Fig. 3 (a).

**Fine-grained structural features** $\{f_{3D}^{(1,i)}\}_{i=1}^N$ are extracted from the source 3D $\hat{S}_{3D}^{\text{src}}$ at a late diffusion timestep $t \approx 0$. Since the denoising network at late timesteps emphasizes structural refinement, these features encode detailed structural information necessary for preserving unedited regions.

**Semantic transition features** $\{f_{3D}^{(2,i)}\}_{i=1}^N$ are derived by conditioning the frozen network on both the source 3D asset $\hat{S}_{3D}^{\text{src}}$ and the target image $I^{\text{tgt}}$ at an early timestep $t \approx 1$. Early denoising stages prioritize semantic alignment with conditioning signals, enabling these features to capture how the structure should evolve to reflect the edit.

Formally, for timesteps $t_1 \approx 0$ and $t_2 \approx 1$, we compute:

$$
\begin{aligned}
\{f_{3D}^{(1,i)}\}_{i=1}^N &= \mathcal{F}(S_{3D}^{\text{src}}, t_1, I^{\text{zero}}), \\
\{f_{3D}^{(2,i)}\}_{i=1}^N &= \mathcal{F}(S_{3D}^{\text{src}}, t_2, I^{\text{tgt}}),
\end{aligned}
\tag{9}
$$

where $\mathcal{F}$ is the frozen Trellis transformer yielding $N$ block-wise features in a single forward pass, $I^{\text{zero}}$ is an empty image condition, and $I^{\text{tgt}}$ is the target edited image.

## 4.3. Time-Adaptive Gating

To balance the contribution of the two feature types throughout denoising, we introduce a time-adaptive gating mechanism, as shown in Fig. 3 (d):

$$(g_1, g_2) = \text{MLP}_{\text{edit}}(t_{\text{embedding}}), \tag{10}$$

generating dynamic weights depending on the timestep embedding $t_{\text{embedding}}$. At early timesteps, the model emphasizes $f_{3D}^{(2,i)}$ to capture semantic transitions, while at later timesteps it prioritizes $f_{3D}^{(1,i)}$ to ensure structural fidelity.

By integrating dual guidance, multi-stage feature extraction, and adaptive gating, 3DEditFormer introduces the first framework that explicitly disentangles *what should change* from *what should remain* in 3D editing. This resolves a fundamental bottleneck of existing methods, providing edits that are localized, consistent, and structure-preserving—an essential step toward scalable 3D editing.

## 4.4. Training and Inference

Our 3DEditFormer follows the two-stage generation paradigm established in Trellis (Xiang et al., 2025). In the first stage, a transformer $\mathcal{T}_{\theta_1}^{(1)}$ generates coarse voxelized shapes that capture the global structure. In the second stage, a separate transformer $\mathcal{T}_{\theta_2}^{(2)}$ refines fine-grained texture and appearance features, which are subsequently decoded into

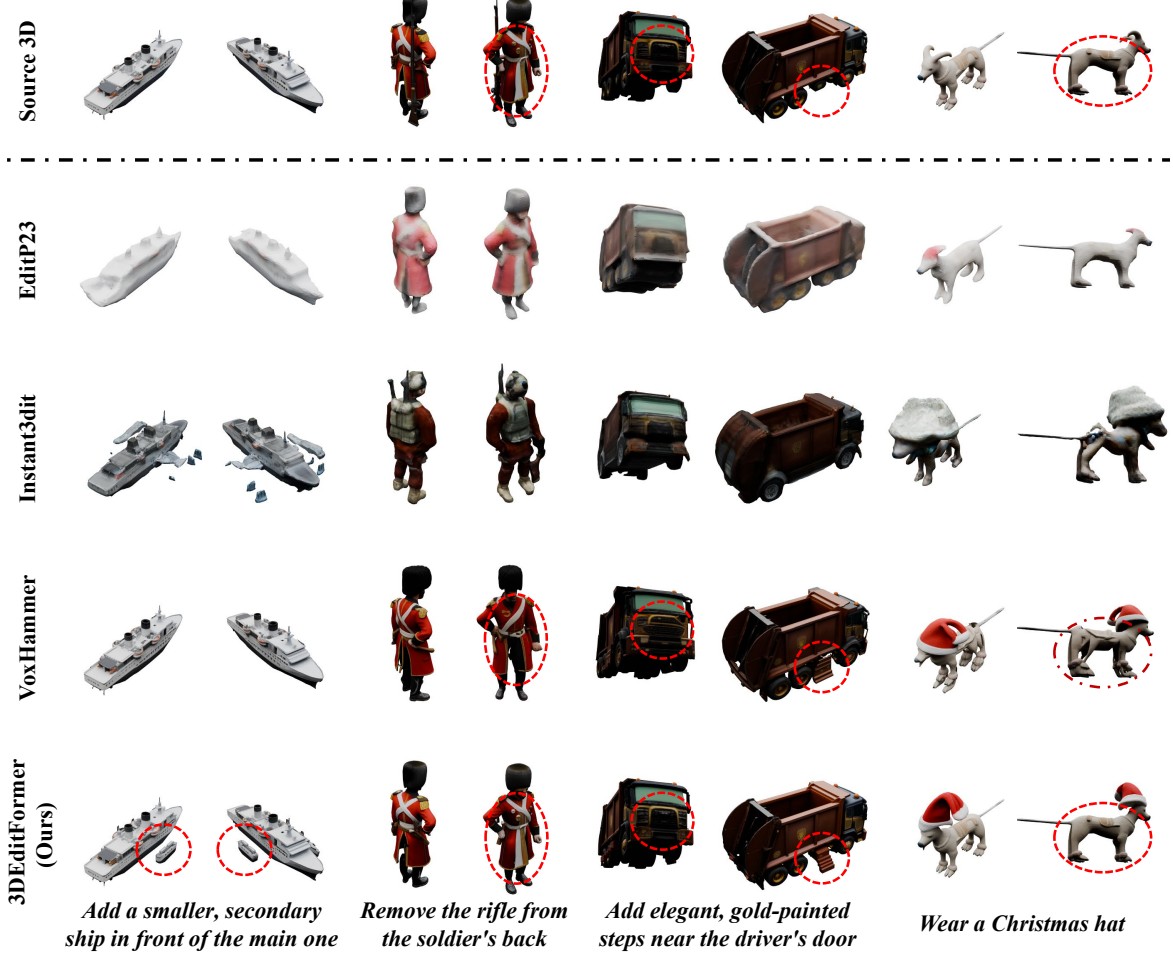

*Figure 4.* Qualitative comparison among our proposed 3DEditFormer and SoTAs, including EditP23 (Bar-On et al., 2025), Instant3dit (Barda et al., 2025), and VoxHammer (Li et al., 2025a) on our proposed 3DEditVerse test set. More visualizations are provided in appendix C (General object editing), D (Character pose editing), E (Non-human pose editing), F (Deformation-like editing), and G (Real-object editing).

explicit 3D representations such as 3D Gaussians (Kerbl et al., 2023) or meshes via a VAE-based decoder (Kingma & Welling, 2013).

The two transformers are parameterized independently but are both trained under the same Conditional Flow Matching (CFM) objective (Lipman et al., 2023):

$$\mathcal{L}(\theta_k) = \mathbb{E}_{t, \boldsymbol{x}_0, \boldsymbol{\epsilon}} \|\mathcal{T}_{\theta_k}^{(k)}(\boldsymbol{x}, t) - (\boldsymbol{\epsilon} - \boldsymbol{x}_0)\|_2^2, \qquad (11)$$

where $\boldsymbol{x}(t) = (1-t)\boldsymbol{x}_0 + t\boldsymbol{\epsilon}$ interpolates between a clean sample $\boldsymbol{x}_0$ and noise $\boldsymbol{\epsilon}$ with timestep $t$. Here, $\mathcal{T}_{\theta_k}^{(k)}$ is either $\mathcal{T}_{\theta_1}^{(1)}$ or $\mathcal{T}_{\theta_2}^{(2)}$ for the corresponding training.

## 5. Experiments

With a frozen Trellis, only 252M parameters are fine-tuned for 40k iterations across the voxel generation

and texture refinement stages with batch size 16 using AdamW (Loshchilov & Hutter, 2017).

For **3D metrics**, we follow (Wu et al., 2024b), and uniformly sample 100,000 points from both the predicted mesh and the ground-truth mesh. **(1)** Chamfer Distance (CD) (Fan et al., 2017) computes the average closest-point distance between the two sets, while **(2)** Normal Consistency (NC) (Gkioxari et al., 2019) measures the alignment of surface normals, capturing geometric fidelity. **(3)** F1$^{0.01}$ (Knapitsch et al., 2017) reports the harmonic mean of precision and recall under a strict distance threshold of 0.01, reflecting preservation of fine geometric details.

For **2D metrics**, following (Li et al., 2025a), each mesh is rendered from 10 fixed camera viewpoints. **(1)** PSNR quantifies pixel-level reconstruction accuracy, and **(2)** SSIM (Wang et al., 2004) evaluates structural similarity

*Table 2.* **Quantitative results on 3DEditVerse under two settings. (a) Full Dataset** benchmarks mask-free methods on all test pairs. **(b) w/o Char-Anim** compares with mask-dependent approaches (Instant3dit, VoxHammer) by excluding samples that lack the required 3D masks. "3D Mask" indicates mask requirement at inference. "Impro." denotes relative improvement over the respective baseline. $\text{VoxHammer}_{+X\%}$ denotes evaluations with ground-truth masks expanded by $X\%$.

*(a)* Comparison on the **Full** 3DEditVerse dataset (1,500 test pairs).

| Method | 3D Mask | 3D Metrics | | | | 2D Metrics | | | | |
|---|---|---|---|---|---|---|---|---|---|---|
| | | CD↓ | NC↑ | F1$^{0.01}$ ↑ | **Impro.**↑ | PSNR↑ | SSIM↑ | LPIPS↓ | DINO-I↑ | **Impro.**↑ |
| EditP23 | ✗ | 46.19 | 0.689 | 32.33 | - | 18.32 | 0.870 | 0.158 | 0.785 | - |
| **3DEditFormer** | ✗ | **13.84** | **0.830** | **64.30** | **63.1%** | **24.40** | **0.918** | **0.068** | **0.963** | **39.5%** |

*(b)* Comparison on the **w/o Char-Anim** subset (excluding character-animation compositions).

| Method | 3D Mask | 3D Metrics | | | | 2D Metrics | | | | |
|---|---|---|---|---|---|---|---|---|---|---|
| | | CD↓ | NC↑ | F1$^{0.01}$ ↑ | **Impro.**↑ | PSNR↑ | SSIM↑ | LPIPS↓ | DINO-I↑ | **Impro.**↑ |
| Instant3dit | ✔ | 29.34 | 0.734 | 32.84 | - | 20.16 | 0.868 | 0.132 | 0.840 | - |
| VoxHammer | ✔ | 9.84 | 0.885 | 77.22 | 74.1% | 26.11 | **0.942** | 0.052 | 0.959 | 37.6% |
| $\text{VoxHammer}_{+9\%}$ | ✔ | 10.27 | 0.880 | 75.56 | 71.7% | 25.83 | 0.939 | 0.055 | 0.958 | 36.2% |
| $\text{VoxHammer}_{+18\%}$ | ✔ | 10.95 | 0.873 | 73.72 | 68.7% | 25.53 | 0.936 | 0.058 | 0.956 | 34.8% |
| $\text{VoxHammer}_{+27\%}$ | ✔ | 11.42 | 0.867 | 72.02 | 66.2% | 25.19 | 0.933 | 0.060 | 0.955 | 33.6% |
| **3DEditFormer** | ✗ | **7.04** | **0.904** | **86.05** | **87.1%** | **26.42** | 0.938 | **0.045** | **0.962** | **39.9%** |

*Table 3.* Ablation study on the effectiveness of the Dual-Guidance Attention Block (Fine-Grained Structural Features $f_{3D}^{(1)}$ + Semantic Transition Features $f_{3D}^{(2)}$) and Time-Adaptive Gating.

| Methods | 3D Metrics | | | 2D Metrics | | | |
|---|---|---|---|---|---|---|---|
| | CD↓ | NC↑ | F1$^{0.01}$ ↑ | PSNR↑ | SSIM↑ | LPIPS↓ | DINO-I↑ |
| Baseline | 16.230 | 0.814 | 60.183 | 23.656 | 0.910 | 0.0784 | 0.956 |
| + Fine-Grained Feat. $f_{3D}^{(1)}$ | 14.586 | 0.825 | 63.701 | 24.021 | 0.913 | 0.0699 | 0.960 |
| + Semantic Feat. $f_{3D}^{(2)}$ | 14.084 | 0.828 | 64.023 | 24.252 | 0.916 | 0.0687 | 0.962 |
| + Time-Adaptive Gating | **13.843** | **0.830** | **64.297** | **24.395** | **0.918** | **0.0682** | **0.963** |

in luminance, contrast, and texture. **(3)** LPIPS (Zhang et al., 2018), based on deep perceptual features, reflects perceptual similarity Finally, **(4)** DINO-I assesses semantic consistency by computing cosine similarity between DINOv2 (Oquab et al., 2024) image embeddings.

### 5.1. Main Results

**Qualitative Comparison.** We present qualitative comparisons with SoTA methods in Fig. 4. One can observe that EditP23 (Bar-On et al., 2025) fails to preserve geometry and texture fidelity, often yielding over-smoothed or incomplete results (e.g., the ship losing structural detail and the soldier's uniform becoming blurred). Instant3dit (Barda et al., 2025) generates edited variants but introduces severe artifacts, such as broken geometry in the ship and collapsed textures in the soldier, indicating instable localized edits.

VoxHammer (Li et al., 2025a) demonstrates stronger geometric fidelity than EditP23 and Instant3dit but is highly sensitive to mask accuracy. When 3D masks are imprecise, its editing consistency deteriorates rapidly, as illustrated by

the red circles in Fig. 4. In contrast, our 3DEditFormer does not require any 3D masks: edits are guided solely by the target image, greatly simplifying the pipeline while preserving both structure and consistency. For instance, it successfully adds a secondary ship without distorting the original vessel and removes the soldier's rifle while maintaining uniform integrity. These results show that 3DEditFormer achieves faithful localized edits and preserves unedited regions, outperforming prior methods in both accuracy and usability.

**Quantitative Comparison.** Tab. 2 reports quantitative results against SoTA methods on our 3DEditVerse test set. Our 3DEditFormer consistently outperforms existing methods across both 3D and 2D metrics. EditP23 (Bar-On et al., 2025) exhibits the weakest performance, with a high CD and low NC, reflecting poor geometric fidelity. Instant3dit (Barda et al., 2025) achieves moderate improvements but suffers from unstable quality, as indicated by low F1$^{0.01}$ and SSIM. VoxHammer (Li et al., 2025a) achieves strong results and the best SSIM when accurate 3D masks are available, highlighting its ability to preserve low-level

structural similarity. However, its reliance on precise masks severely limits its practicality: with even small perturbations (e.g., increasing the 3D masks by 9%, 18%, or 27%), Vox-Hammer suffers from severe performance degradation, as shown in Tab. 2. This sensitivity severely limits its practical utility compared to our robust, mask-free approach. Further sensitivity analysis of mask-based methods and comprehensive evaluations on additional metrics are provided in appendix I and J, respectively.

In contrast, our 3DEditFormer attains the best overall performance without any auxiliary 3D mask, achieving a +13% improvement of 3D Metrics than VoxHammer. It surpasses prior methods on CD, NC, F1, PSNR, LPIPS, and DINO-I, while remaining competitive on SSIM. These results demonstrate that 3DEditFormer achieves SoTA and consistency through a simpler, more robust pipeline, removing the need for external mask supervision.

### 5.2. Ablation Study

Tab. 3 summarizes the contribution of each 3DEditFormer component. Compared to the baseline that uses vanilla cross-attention with $S_{3D}^{\text{src}}$ but lacks multi-stage guidance, incorporating fine-grained structural features $f_{3D}^{(1)}$ significantly boosts CD, NC, and F1, validating their role in preserving unedited geometric details.

Building on this, the addition of semantic transition features $f_{3D}^{(2)}$ further enhances both 2D and 3D metrics. These early-stage features provide complementary cues that effectively guide structural adaptation toward the intended target edits.

Finally, incorporating the proposed time-adaptive gating mechanism delivers the strongest overall performance. By dynamically balancing $f_{3D}^{(1)}$ and $f_{3D}^{(2)}$ across denoising steps, the model ensures high-quality, localized edits while maintaining rigorous structural integrity. Further analysis of this mechanism is provided in Appendix K.

## 6. Conclusion

We introduce 3DEditVerse, the first large-scale benchmark for paired 3D editing, containing about 118K asset pairs with diverse geometry and appearance edits, designed for scalability and semantic consistency. Then, we propose 3DEditFormer, a 3D-structure-preserving transformer with dual-guidance attention and time-adaptive gating, enabling precise and consistent 3D edits without 3D masks. Extensive experiments demonstrate SoTA performance, achieving a strong balance between fidelity and practicality.

**Limitation Discussion.** Our 3DEditFormer relies on latent-space editing, which, while efficient, may introduce precision loss when handling high-resolution 3D assets. Fine geometric details may be degraded during latent transfor-

mation. Future work could explore editing directly in the vanilla 3D domain to preserve fine-grained fidelity better.

## Acknowledgements

This work was supported by the Singapore Ministry of Education (MOE) Academic Research Fund (AcRF) Tier 1 grant (Proposal ID: 23-SIS-SMU-070), the National Natural Science Foundation of China under Grant 62233005, Grant U25B6002, Grant U2441245 and the Fundamental Research Funds for the Central Universities. Any opinions, findings and conclusions or recommendations expressed in this material are those of the author(s) and do not reflect the views of the Ministry of Education, Singapore.

## Impact Statement

This paper presents work whose goal is to advance the field of 3D Editing. There are many potential societal consequences of our work, none which we feel must be specifically highlighted here.

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

# A. Visualization of our 3DEditVerse dataset

## (a) Character–Animation Compositions

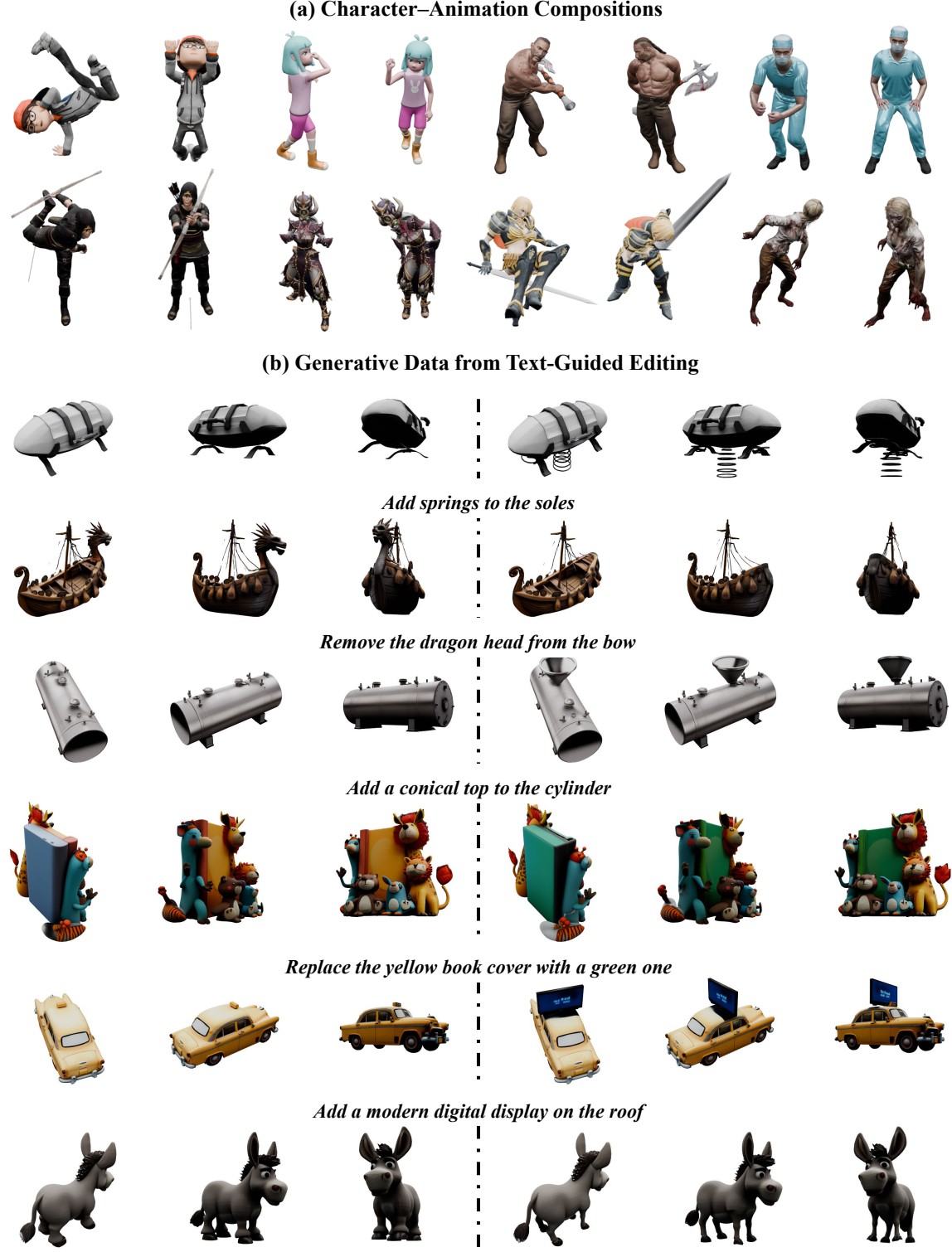

## (b) Generative Data from Text-Guided Editing

*Add springs to the soles*

*Remove the dragon head from the bow*

*Add a conical top to the cylinder*

*Replace the yellow book cover with a green one*

*Add a modern digital display on the roof*

*Replace the donkey's front legs with longer, more slender ones*

*Figure 5.* More examples of (a) Character–Animation Compositions and (b) generative data from text-guided editing in our proposed 3DEditVerse dataset.

# B. Details of instructional prompts

## B.1. Prompt Example of DeepSeek-R1 for Vocabulary to Text

---

Prompt Example of DeepSeek-R1 for Vocabulary to Text

**Instruction:** You are an expert creative assistant for a text-to-image AI pipeline. Your task is to generate descriptive text prompts for creating an initial image.
Specifically, for the object "plane":
Generate a numbered list of 20 text prompts (around 20 words each). Each prompt should contain some attributes or adjectives that describe the object.
**Constraint:** Do not describe any background or environment or action, as the object will be rendered on a pure white background.
**Output Format:**
Text Prompts:
1. [Text Prompt 1]
2. ...
20. [Text Prompt 20]

**Output:** Text Prompts:
1. A sleek silver jet plane with swept-back wings and twin rear engines on a white background.
2. A vintage propeller plane featuring rounded wings and a radial engine in polished chrome.
3. A military stealth plane with angular edges and matte black coating, no visible seams.
4. A small white private plane with high-mounted wings and a single nose propeller.
5. A cargo plane showcasing a boxy fuselage and high tail section with four turbofans.
6. A paper airplane with sharp creases and pointed wings folded from blue paper.
7. A futuristic concept plane with smooth curves and retractable wingtips in metallic teal.
8. A seaplane with large pontoons and reinforced wing struts in faded yellow.
9. A biplane with dual red fabric wings and exposed cockpit wires.
10. A toy plane with chunky plastic wings and oversized propellers in primary colors.
11. A drone plane featuring multiple rotors and a compact carbon-fiber body.
12. A glider plane with extra-long tapered wings and minimalist cockpit framing.
13. A fighter jet with delta wings and afterburner nozzles in camouflage gray.
14. A crop duster plane with chemical tanks and sturdy landing gear in olive green.
15. A stunt plane with checkerboard patterns and reinforced wing supports.
16. A commercial airliner with elongated body and blended winglets in navy blue.
17. A spaceplane with heat-shield tiles and retractable landing legs.
18. A flying wing design with no tail section and embedded engines in charcoal gray.
19. A micro-light plane with fabric-covered frame and open cockpit in bright orange.
20. A turbo-prop plane with wing-mounted engines and triple-blade propellers.

---

In the above example, we illustrate the case of the vocabulary "plane", where the prompt provided to DeepSeek-R1 (Guo et al., 2025) and its corresponding outputs are shown. Specifically, for each vocabulary word, we generate 20 diverse textual descriptions to enhance edit diversity.

## B.2. Prompt Example of Flux.1-Dev for Source Image Generation

---

Prompt Example of Flux.1-Dev for Source Image Generation

**Instruction:** "A glider plane with extra-long tapered wings and minimalist cockpit framing" in the form of an object in the middle of a pure white background, no shadow

---

For each textual description, we append a specific template that places the object at the center of a pure white background, ensuring a clean and unambiguous layout generated by Flux.1-Dev (Labs, 2024). This design facilitates reliable image-to-3D lifting in subsequent stages. The resulting source images are illustrated in Fig. 6.

## B.3. Prompt Example of Qwen-VL for Edit Prompt Generation

---

Prompt Example of Qwen-VL for Edit Prompt Generation

**Instruction:** You are an expert visual analyst for an advanced image editing pipeline. Your task is to analyze the provided image of an object and generate high-quality editing prompts.
**\*\*Analysis & Goal:\*\***
Look closely at the object in the image. Your goal is to suggest a list of 5 specific \*\*substantial, and obvious shape transformations\*\*. These edits will be used by another AI to modify this exact image.
**\*\*Core Instructions:\*\***
Each suggested edit prompt must fall into one of the following three categories:
1. \*\*Add a Part: Propose adding a new, structurally significant component that is logically related to the main object.
\* \*Example based on a cup image: "Add a thick, rectangular handle to its side."
2. \*\*Remove a Part: Identify a visually distinct and major part of the object and propose its removal.
\* \*Example based on a chair image: "Remove the entire backrest."
3. \*\*Replace a Part: Identify a specific part of the object and propose replacing it with a new part of a distinctly different shape.
\* \*Example based on a lamp image: "Replace its slender cylindrical base with a wide pyramid base."
**\*\*Strict Constraints:\*\***
\* \*\*Focus ONLY on Shape:\*\* Do not suggest changes related to color, texture, material (e.g., "make it wooden"), or lighting. The edits must alter the object's geometry.
\* \*\*Be Specific:\*\* Refer to parts you can actually see in the image (e.g., "the base," "the top," "the legs," "the handle").

**Output:** 1. Remove the vertical stabilizer from the tail section.
2. Add a small, rounded winglet to the tip of each wing.
3. Replace the nose cone with a more pointed, aerodynamic design.
4. Add a set of flaps to the trailing edge of the wings.
5. Replace the horizontal stabilizers with larger, swept-back surfaces.

---

We feed each generated source image into the Qwen2.5-VL-7B model (Bai et al., 2025) to produce diverse editing instructions. The example above corresponds to the airplane image shown in the top-left corner of Fig. 6. Each resulting edit prompt is then paired with its source image and passed to the Flux.1-Kontext model (Labs et al., 2025) for image editing. Figure 6 illustrates several examples of edited results alongside their corresponding prompts.

## B.4. Prompt Example of Qwen-VL for Edit Region Localization

---

Prompt Example of Qwen-VL for Edit Prompt Generation

**Instruction:** Given the original image and the edited image concatenated side-by-side:
A text instruction describes the editing area: "Remove the canopy above the pump"
Your task:
1. Compare the two images and locate only the regions in the left image corresponding to the given editing area: "canopy"
2. 2. Output the bounding box(es) for the regions in the format: [[xmin, ymin, xmax, ymax]] (values in pixels, only in the 512x512 left image).
3. If you cannot find the edited region (i.e. "canopy"), output an empty list: []
\*\*Directly output the bounding box(es)\*\*

**Output:** [103, 52, 396, 104]

---

We concatenate the source and target images side by side and provide them to the Qwen-VL-Max model (Bai et al., 2025) together with the above prompt, which enables the model to localize the edited region by predicting bounding boxes corresponding to the modification.

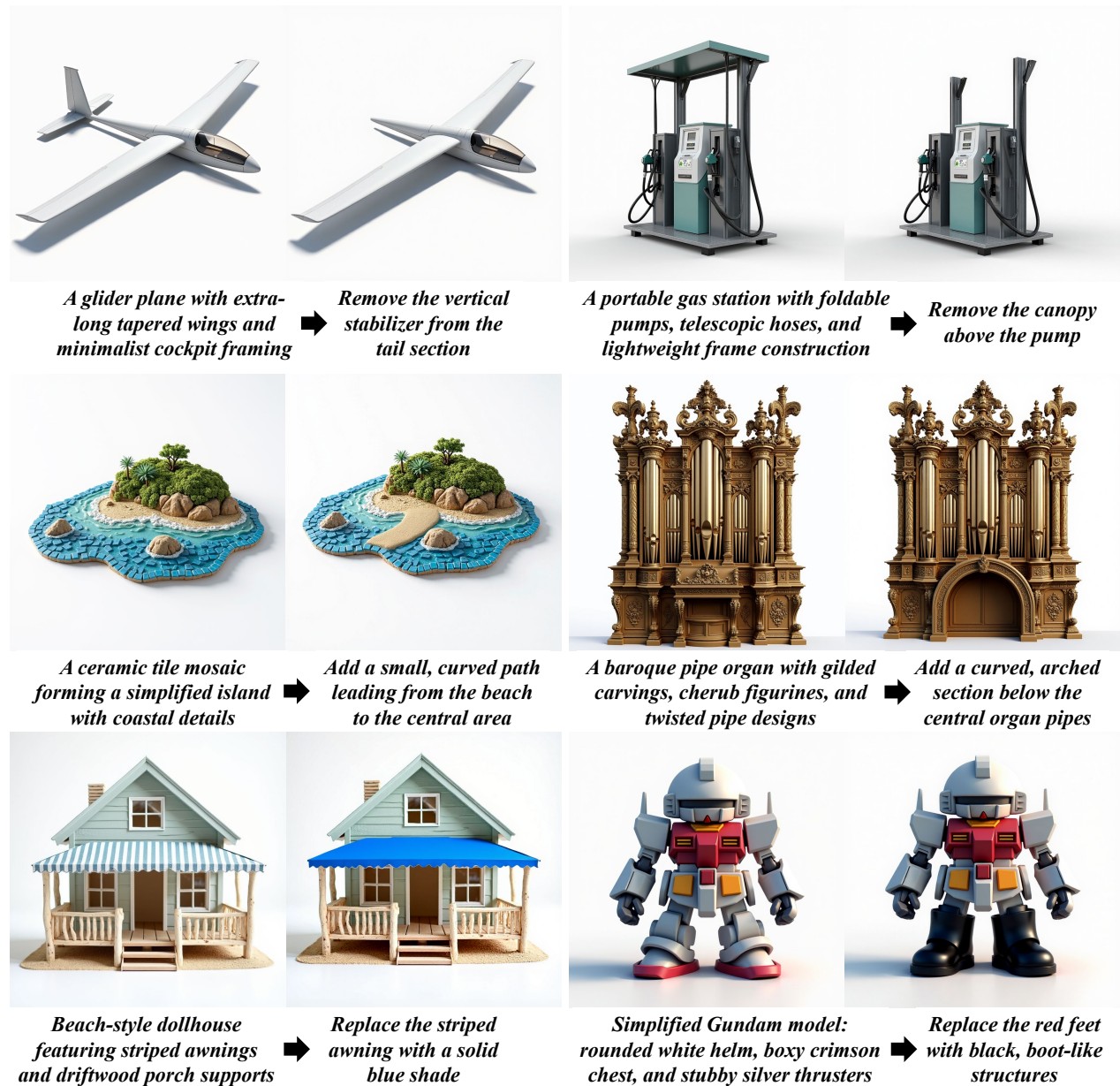

*Figure 6.* Examples of source–target image pairs. Text below the source (left) shows the generation prompt, while text below the target (right) shows the editing instruction.

## C. Visualization of Comparison with SoTA Methods

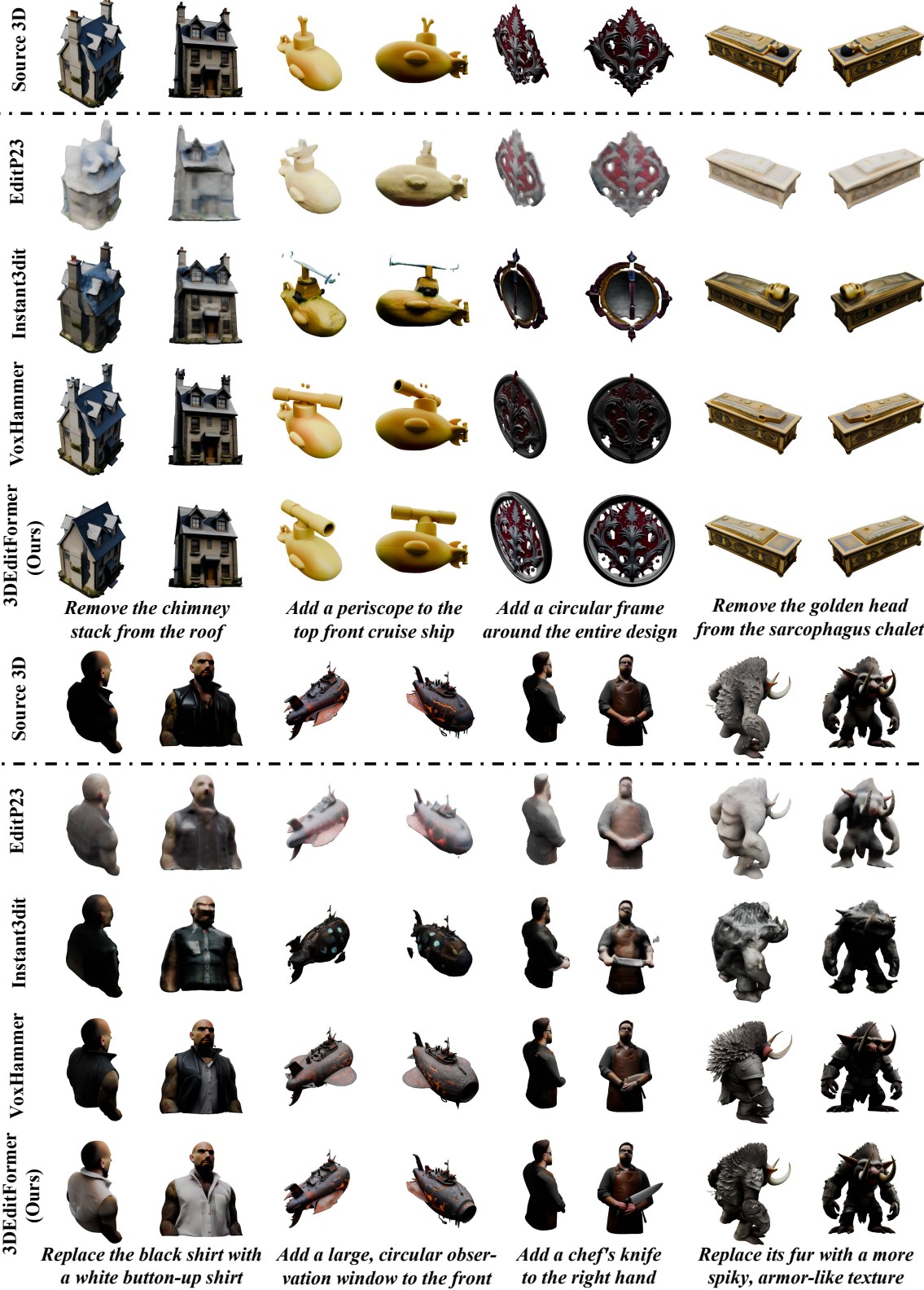

*Figure 7.* More qualitative results compared with SoTA methods.

## D. Visualization on Character–Animation Test Set

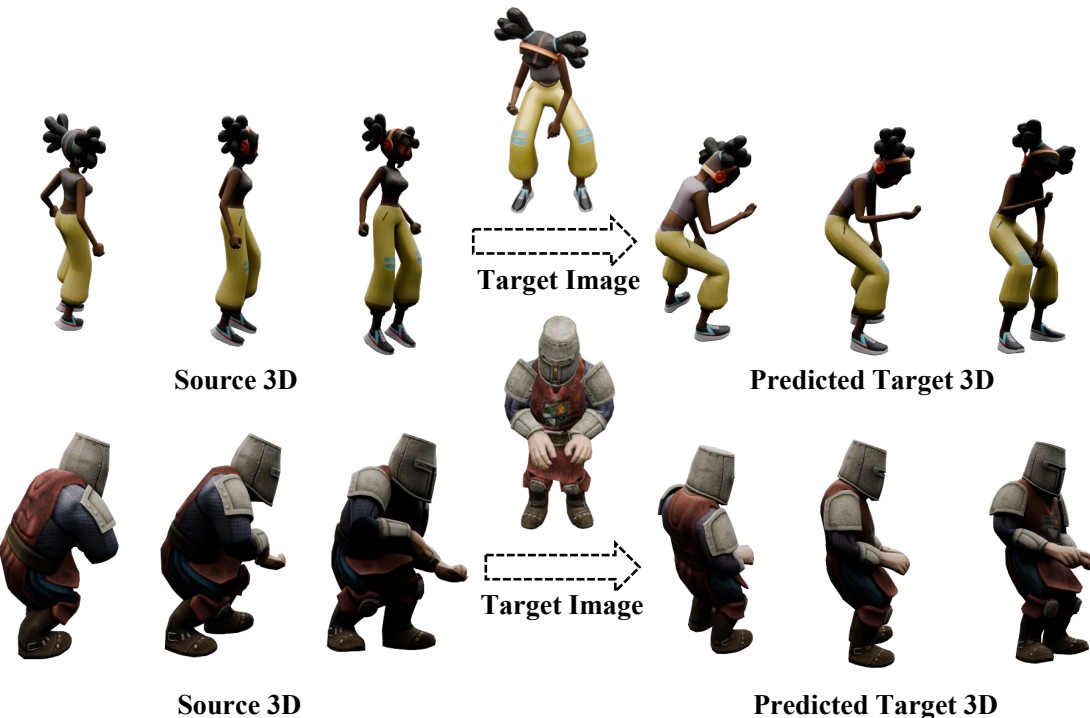

*Figure 8.* Qualitative results of our 3DEditFormer on character–animation test set.

Fig. 8 presents qualitative results of our 3DEditFormer on the Character–Animation Test Set. Given a source 3D asset and a target image depicting a new pose, our method successfully generates target 3D assets that accurately capture the articulated geometry and maintain texture fidelity. The results demonstrate that 3DEditFormer is able to produce realistic, pose-driven edits while preserving consistency across views, highlighting its effectiveness for complex character–animation scenarios.

## E. Visualization on Non-Human Pose Editing

To demonstrate the generalization capability of 3DEditFormer beyond standard human characters, we present results on non-human articulated objects. Unlike human bodies which often share standard topologies, these objects require the model to infer diverse semantic parts without explicit skeletal supervision or category-specific priors. As illustrated in Fig. 9, the model successfully executes complex articulation commands, such as identifying the neck joint to naturally "raise the deer's head", and unfolding wings from a resting state to "make the eagle fly." Additionally, it performs precise localized limb manipulation to "raise the monkey's left hand," confirming that our approach effectively generalizes to diverse kinematic structures beyond human datasets.

## F. Visualization on Deformation-Like Editing

Beyond rigid articulation, our framework proves robust in handling continuous geometric deformations that involve global or semi-global changes to object proportions and curvature. 3DEditFormer achieves this by manipulating shape descriptors in the latent space to stretch, compress, or straighten geometry without introducing artifacts or breaking topological connectivity. As shown in Fig. 10, when instructed to "make the dog's limbs shorter and its body plumper," the model performs a coordinated multi-axis deformation to achieve a stylized proportion. Similarly, for the instruction "make the tree grow straight," it effectively corrects the global curvature of the trunk vertically, demonstrating the capability to perform structural rectification while preserving fine-grained texture and branch details.

**Source 3D** | **Edited Results**

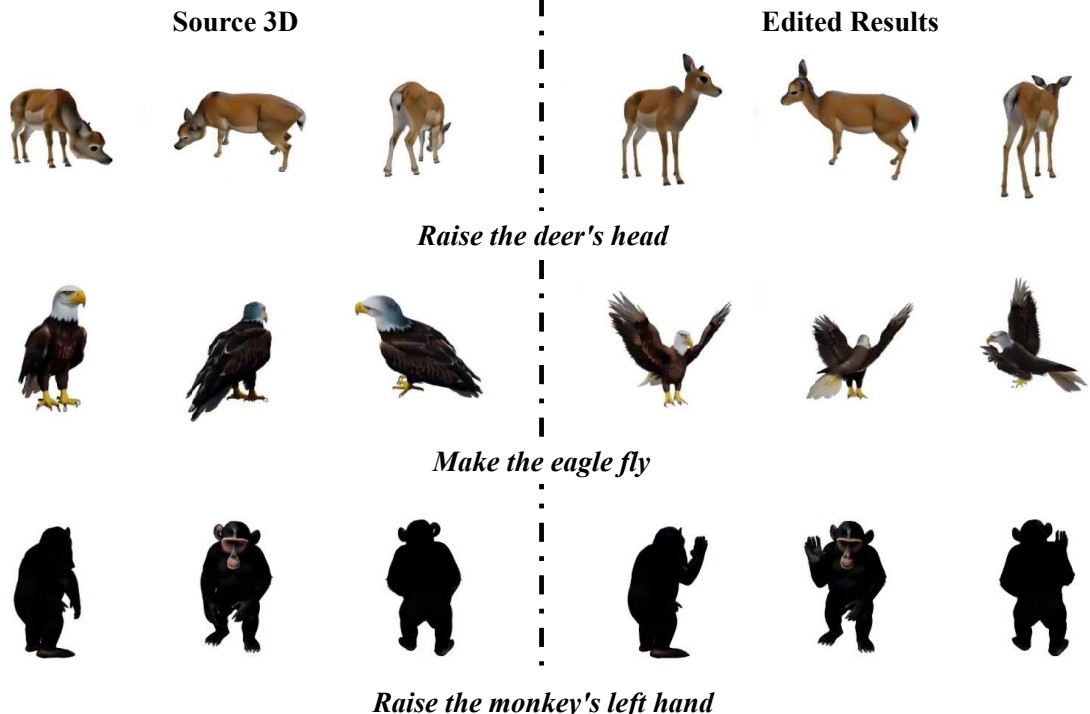

*Raise the deer's head*

*Make the eagle fly*

*Raise the monkey's left hand*

*Figure 9.* Qualitative results of our 3DEditFormer on non-human pose editing.

**Source 3D** | **Edited Results**

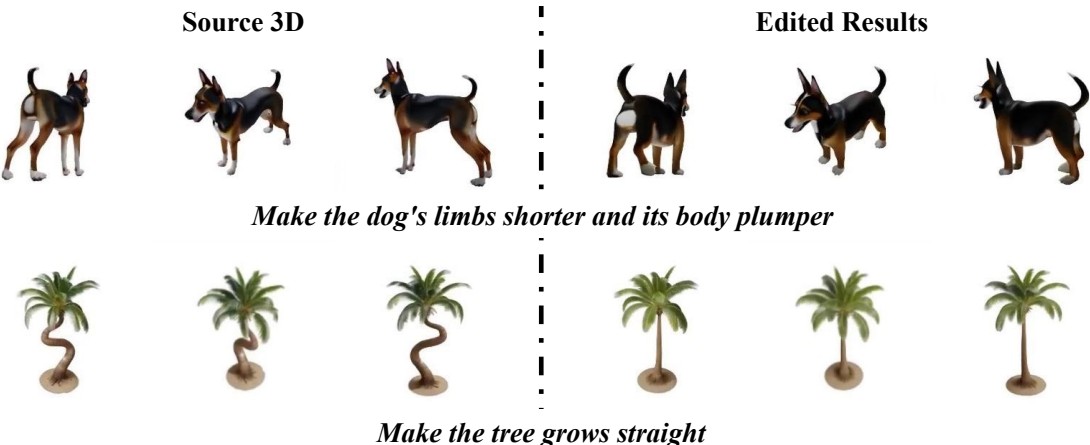

*Make the dog's limbs shorter and its body plumper*

*Make the tree grows straight*

*Figure 10.* Qualitative results of our 3DEditFormer on deformation-like editing.

## G. Visualization on Real-Object Editing

We further extend our evaluation to real-world objects in Fig. 11 to assess robustness against complex topologies and noisy inputs common in non-synthetic data. These scenarios showcase the model's ability to handle drastic topological changes, such as reducing a large basin to a compact structure when "replacing the bathtub with a sink," or managing complex topological growth when transforming rigid crab legs into fluid tentacles to "replace the crab with an octopus." Furthermore, the model demonstrates capable high-frequency detail synthesis, generating intricate mechanical fingers in the "arm to hand holding a gun" case, and synthesizing organic features like teeth and horns when "replacing a sci-fi turret with a dragon head." Finally, the "ice cream to watermelon base" example highlights the ability to maintain texture consistency even under significant self-occlusion during the editing process.

| Source 3D | Edited Results |
|---|---|

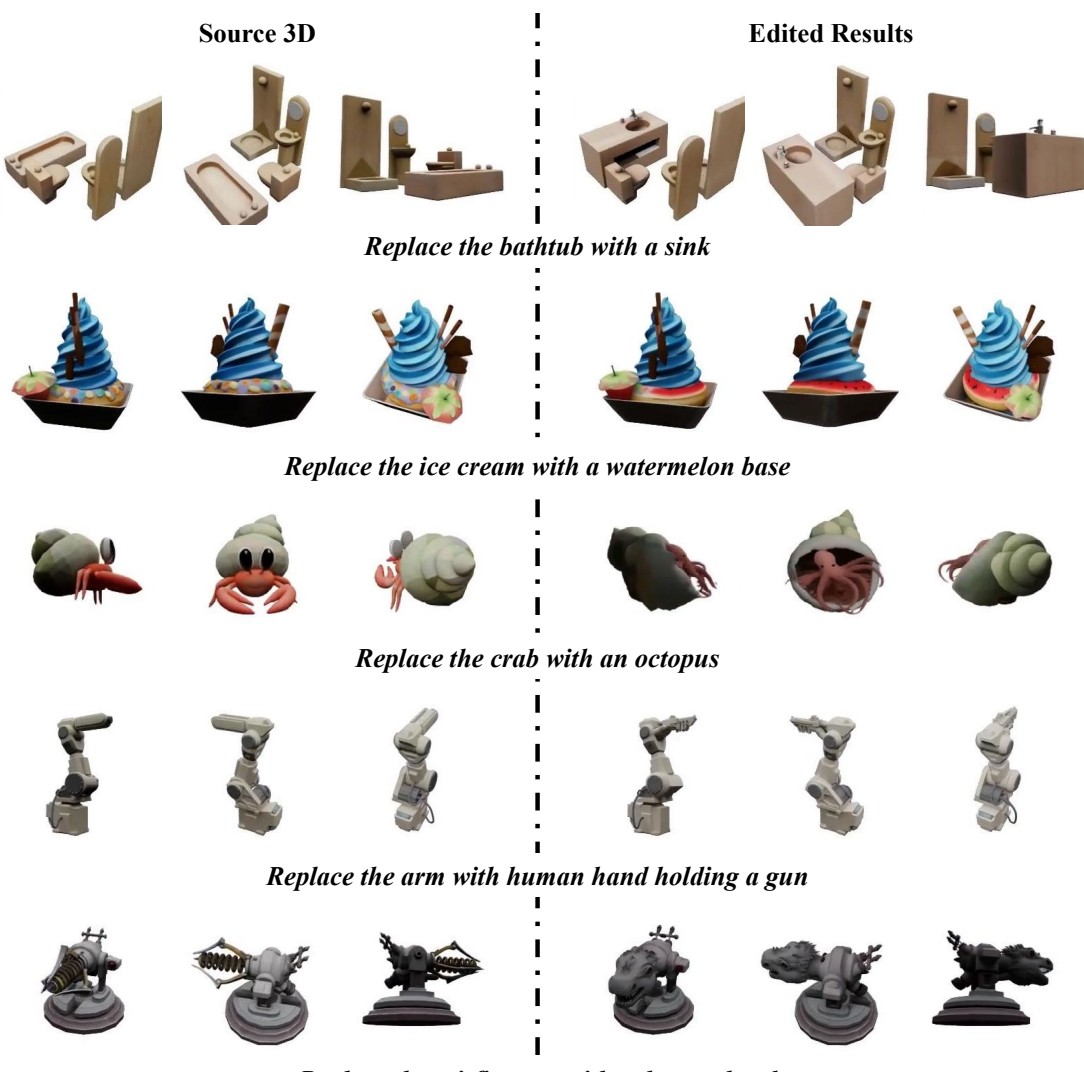

*Replace the bathtub with a sink*

*Replace the ice cream with a watermelon base*

*Replace the crab with an octopus*

*Replace the arm with human hand holding a gun*

*Replace the sci-fi turret with a dragon head*

*Figure 11.* Qualitative results of our 3DEditFormer on real-object editing.

## H. Infeasibility of Mask-Based Methods for Pose-Driven Editing

To further justify the exclusion of mask-based methods (e.g., VoxHammer) from the Character-Animation subset, we visualize the cases of such methods under global pose changes in Fig. 12. In pose-driven editing scenarios, such as a knight falling or crouching (see input columns), the transformation is global rather than local. Consequently, a static 3D mask cannot isolate a specific "editing region" while keeping the rest fixed. To adapt VoxHammer for this task, we defined the editing mask as the entire voxel space, allowing the model to update all voxels.

As illustrated in the comparison, (1) VoxHammer (Right-Top): Lacking a static unmasked region to provide structural priors or anchors, the model fails to maintain geometric integrity. The resulting meshes exhibit severe artifacts, broken topology, and floating noise, demonstrating that the method collapses when the edit requires global deformation rather than local inpainting. (2) 3DEditFormer (Right-Bottom): Our method, which does not rely on explicit 3D masks, effectively utilizes the target image as a global guide. It successfully reconstructs the complex new poses (e.g., the prone and crouching positions) while preserving the semantic identity of the character.

These results confirm that mask-based comparisons on pose-driven data are technically invalid, as the requirement for a static mask is incompatible with global articulation.

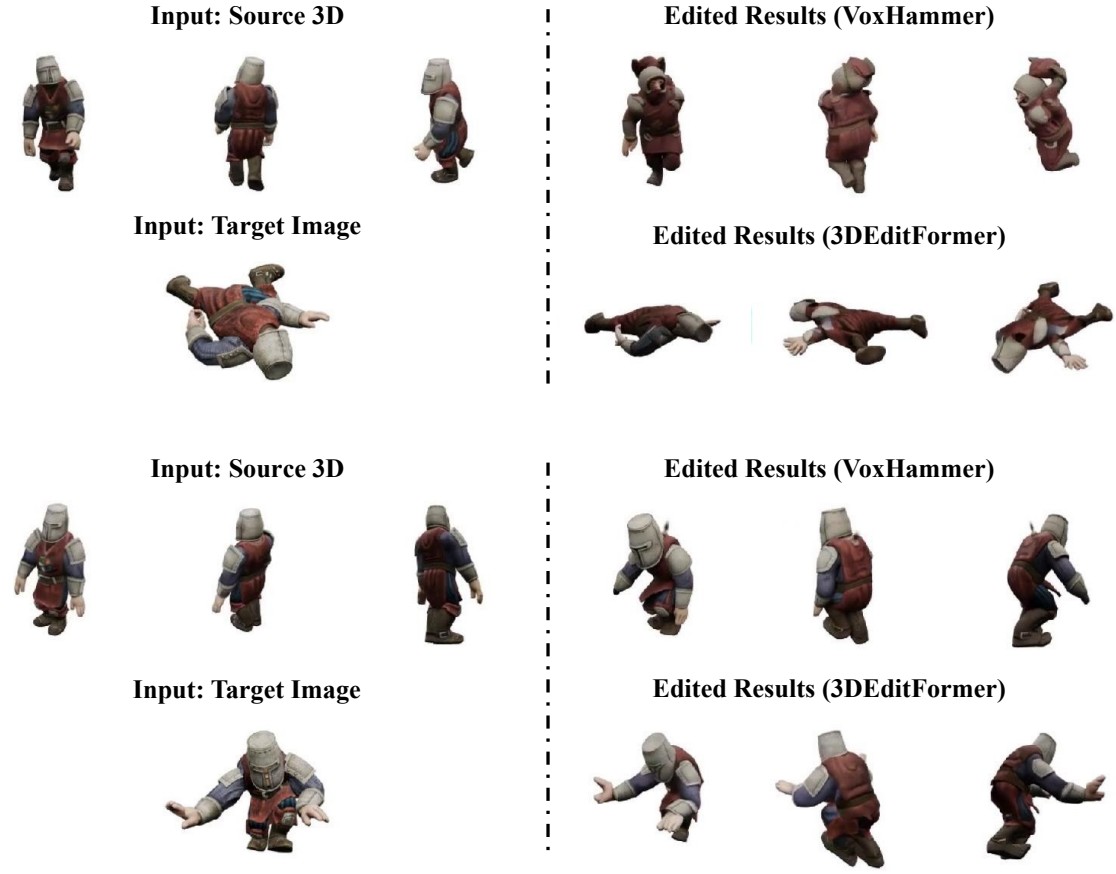

*Figure 12.* Comparison of VoxHammer ([Li et al., 2025a](#)) vs. our 3DEditFormer on pose editing.

## I. Sensitivity Analysis of Mask-Based Methods

To demonstrate the brittleness of methods relying on explicit 3D masks, we visualize the impact of mask precision on VoxHammer in Fig. 13. In real-world scenarios, perfect 3D masks are difficult to obtain; therefore, we simulate practical imperfections by inflating the ground-truth mask radius by 9%, 18%, and 27%.

Fig. 13 presents a visual comparison across three editing prompts. As the mask radius increases, VoxHammer erroneously edits the surrounding environment included in the mask. Comparison with the bottom row (Ours) highlights that 3DEditFormer achieves precise, localized edits (e.g., cleanly modifying the chair legs or the book cover) solely through image guidance. This confirms that our mask-free approach is significantly more robust to the inevitable imprecision of real-world inputs.

## J. Additional Quantitative Evaluations

To provide a more comprehensive evaluation of our proposed 3DEditFormer, we introduce four additional quantitative metrics focusing on two crucial dimensions: high-frequency geometric detail preservation (Laplacian Error and Curvature Error) and text-semantic alignment (Directional CLIP Score and VLM-based Score).

- **Laplacian Error**: This metric captures the difference in differential coordinates between the predicted and ground-truth meshes, which is highly sensitive to unintended smoothing effects. It is formulated as:

$$L(x_i) = x_i - \frac{1}{|N_i|} \sum_{j \in N_i} x_j, \tag{12}$$

where $N_i$ denotes the 1-ring neighborhood of vertex $x_i$, and $|N_i|$ is the number of neighbors. It directly reflects local

surface bending.

- **Curvature Error**: We measure the Root Mean Square Error (RMSE) of the Mean Curvature. This metric identifies whether sharp edges or intricate features are lost during the editing process, as these regions naturally correspond to high-curvature variations.

- **Directional CLIP Score**: This evaluates whether the semantic change between the source and edited 3D assets aligns with the modification vector specified in the text prompt.

- **VLM-based Score**: We leverage a SoTA Vision-Language Model (Qwen3-VL-Plus) (Yang et al., 2025) to provide a holistic, automated score evaluating how faithfully the final 3D asset adheres to the provided editing instructions.

As reported in Table 4, our 3DEditFormer substantially outperforms all baseline methods across all four metrics. Specifically, it achieves the lowest Laplacian and curvature errors, confirming its superior capability to preserve high-frequency geometric integrity and prevent oversmoothing. Simultaneously, our method yields the highest Directional CLIP and VLM scores, validating its exceptional semantic fidelity and text-conformance.

*Table 4.* Quantitative comparison on high-frequency geometric preservation (Laplacian and Curvature errors) and semantic alignment (Directional CLIP and VLM scores).

| Method | Geometric Fidelity | | Semantic Alignment | |
|---|---|---|---|---|
| | Laplacian $(10^{-3})\downarrow$ | Curvature $(10^{-3})\downarrow$ | Directional CLIP $\uparrow$ | VLM Score $\uparrow$ |
| EditP23 | 14.94 | 5.62 | 0.056 | 3.09 |
| Instant3dit | 18.30 | 4.88 | 0.075 | 3.92 |
| VoxHammer | 10.19 | 4.31 | 0.107 | 6.01 |
| **3DEditFormer** | **8.82** | **4.09** | **0.113** | **6.16** |

## K. Robustness of Time-Adaptive Gating (TAG)

To further investigate the stability and generalization of our proposed TAG mechanism, we categorize the test set into three distinct subsets: Texture-Only, Geometry-Only, and Mixed Editing. We then evaluate the model performance with and without TAG across these categories. As shown in Tab. 5, the TAG-enabled model consistently outperforms the baseline across all subsets on both 3D and 2D metrics. Notably, the improvement is uniform, enhancing geometric fidelity in geometry-heavy edits while maintaining high perceptual quality in texture-focused tasks, indicating that TAG does not bias the model toward any specific edit type.

*Table 5.* Comparison of editing performance with and without Time-Adaptive Gating (TAG) across mixed, texture, and geometry edits.

| Editing Type | Methods | 3D Metrics | | | 2D Metrics | | | |
|---|---|---|---|---|---|---|---|---|
| | | CD$\downarrow$ | NC$\uparrow$ | F1$^{0.01}$ $\uparrow$ | PSNR$\uparrow$ | SSIM$\uparrow$ | LPIPS$\downarrow$ | DINO-I$\uparrow$ |
| **Mixed** | w/o TAG | 7.558 | 0.895 | 84.681 | 26.003 | 0.934 | 0.0479 | 0.958 |
| | w/ TAG | **6.965** | **0.902** | **86.247** | **26.408** | **0.939** | **0.0442** | **0.962** |
| **Texture** | w/o TAG | 6.385 | 0.922 | 85.408 | 26.122 | 0.929 | 0.0474 | 0.955 |
| | w/ TAG | **6.160** | **0.924** | **85.804** | **26.131** | **0.932** | **0.0469** | **0.958** |
| **Geometry** | w/o TAG | 8.488 | 0.897 | 83.837 | 26.023 | 0.934 | 0.0501 | 0.960 |
| | w/ TAG | **7.375** | **0.905** | **85.623** | **26.503** | **0.938** | **0.0458** | **0.963** |

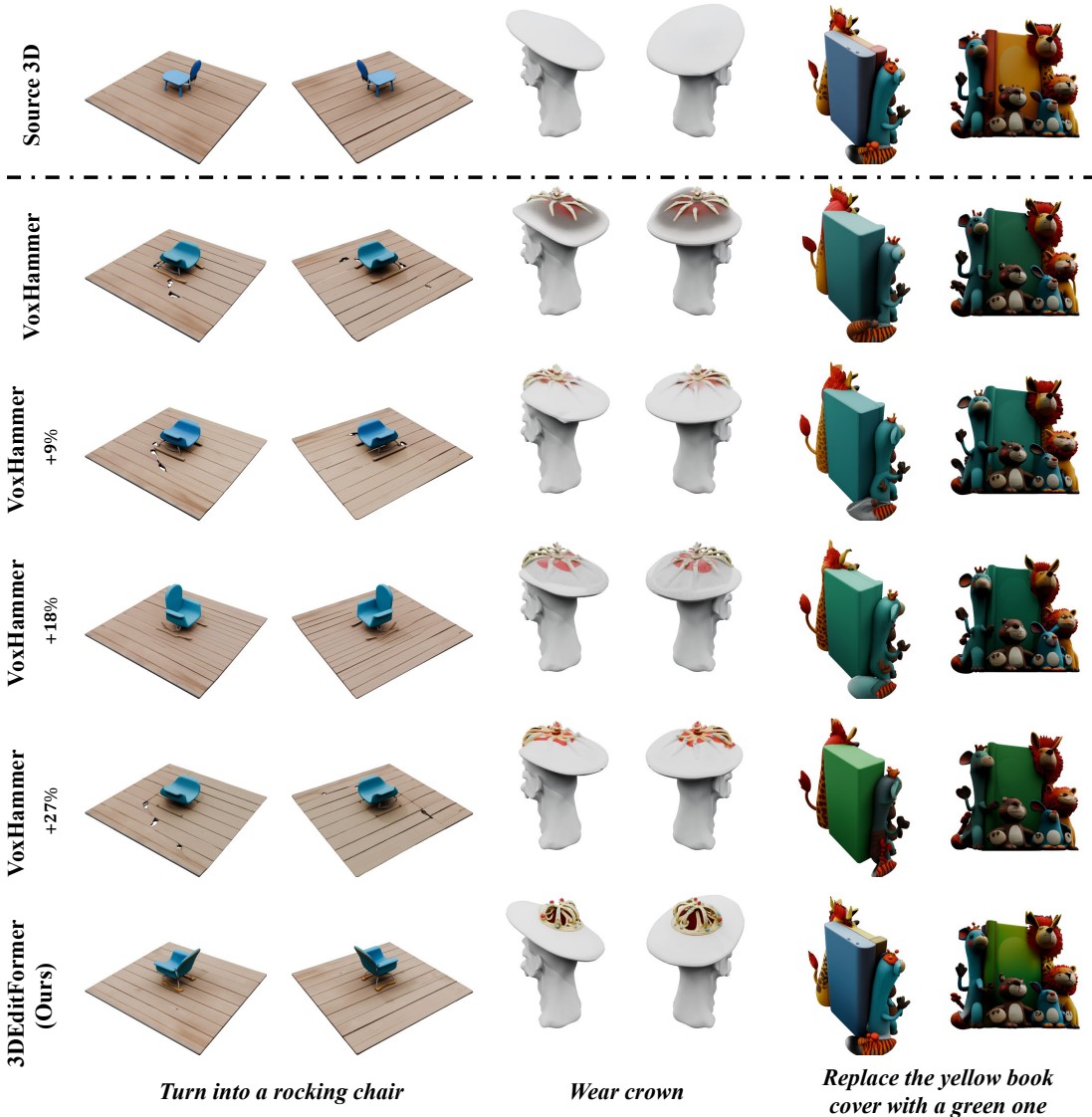

*Figure 13.* Robustness analysis against mask precision. We compare VoxHammer using ground-truth masks vs. inflated masks (+9%, +18%, +27%).

