# OpenReview forum: "Towards Scalable and Consistent 3D Editing"
_ICML.cc/2026/Conference — ICML 2026 regular_

### Official Review · Reviewer_bcYR · 2026-03-09

**Soundness:** 3
**Presentation:** 3
**Significance:** 3
**Originality:** 3
**Overall Recommendation:** 4
**Confidence:** 3

**Summary:**

This paper introduces a dataset and a method for 3D editing based on large-scale foundation models. By leveraging language models and 3D generative models, the authors collect 116,309 paired data samples. They propose a model built upon Trellis, enhanced by a dual attention mechanism. Comparative results demonstrate superior performance in 3D editing tasks.

**Compliance With Llm Reviewing Policy:**

Affirmed.

**Key Questions For Authors:**

See weakness/

**Limitations:**

Yes

**Strengths And Weaknesses:**

-Regarding the data generation pipeline, aside from using DINOv2 for consistency checking, does the process involve any human validation? It remains challenging to guarantee data quality through a fully automatic pipeline alone.

-The proposed model is trained on the newly constructed dataset, where all data are generated using Trellis. However, it appears that the compared baseline methods did not have access to this dataset during evaluation. This raises concerns about the fairness of the experimental comparison.

-It is not clear about the technical contribution due to the common dual attention mechnism in existing works.

---

> ### Author Rebuttal · Authors · 2026-03-30
>
> **To Reviewer bcYR**
>
> We appreciate your insightful and constructive feedback. In the following, we provide our point-by-point response and hope our response helps address your concerns. We also look forward to the subsequent discussion which further helps to solve the current issues.
>
> > **Q1: Data quality assurance in the dataset generation pipeline.**
>
> **A1:** We acknowledge the risk of noise in automated pipelines and thus have implemented several mechanisms to ensure data quality:
>
> *  **Robust 3D Masking**: We employ a multi-view voting strategy where 2D masks from 70 viewpoints are back-projected into 3D space. A voxel is only retained if consistently supported by multiple views, effectively filtering out random segmentation noise.
>
> *  **Automated Quality Filtering**: After 3D lifting, we render the edited asset and calculate DINOv2 feature similarity against the target image. Samples falling below a strict threshold are automatically discarded to remove artifacts and identity mismatches.
>
> *  **Manual Test Set Curation**: While the training set is automated for scale, the 1,500 test pairs were rigorously manually evaluated and curated to guarantee a reliable evaluation benchmark.
>
> The effectiveness of these quality-assurance mechanisms is validated by the superior quality of the resulting benchmark and its significant impact on model performance:
>
> * **Superior Dataset Quality**: 3DEditVerse is the largest high-quality benchmark to date, featuring 116,309 training pairs. Unlike prior datasets such as CMD or 3D-Alpaca-Editing, it is the first to simultaneously achieve large-scale scalability, multi-view consistency, and semantic harmony.
>
> * **Proven Data Effectiveness**: The high quality of our data is directly reflected in the model's performance. 3DEditFormer trained on this dataset achieves SoTA results, including a 13% average improvement in 3D metrics over VoxHammer without requiring any manual 3D masks. This demonstrates that our automated pipeline provides a robust and high-fidelity supervision signal.
>
> While our current automated pipeline proves highly effective, we agree that **incorporating a human-in-the-loop process for the large-scale training set is a valuable direction for future work to further enhance data purity**.
>
> > **Q2: Fairness of comparison regarding the use of the newly constructed dataset.**
>
> **A2:** This is a crucial point, and we appreciate the opportunity to clarify. Our experimental goal is not to compare model weights trained on the same data, but to compare fundamentally different **editing paradigms**:
>
> * **Training-Free vs. Trained Paradigms:** SoTA methods like **VoxHammer** is designed to be **training-free**, meaning they do not require a training dataset to function. Therefore, providing them with training data is not applicable to their architecture.
> * **Mask-Required vs. Mask-Free:** A primary contribution of our work is achieving precise 3D editing **without explicit 3D masks**. While VoxHammer relies on manual or auxiliary 3D masks to anchor its edits, our **3DEditFormer** learns to disentangle editable regions through supervised training on 3DEditVerse.
>
> Our results demonstrate that our **trained, mask-free paradigm** not only offers a more practical workflow but also achieves superior fidelity compared to the **training-free, mask-required paradigm**, even when the latter is supplied with 3D masks. This finding robustly supports the advantages of our proposed approach.
>
> > **Q3: Technical contribution and novelty of the dual attention mechanism.**
>
> **A3:** Our "Dual-Guidance Attention" is novel and fundamentally different from prior "Dual Attention" frameworks (e.g., ECCV22--*DaViT: Dual Attention Vision Transformers* or CVPR19--*Dual Attention Network for Scene Segmentation*) in both motivation and architecture:
>
> * **Focus (Temporal Dynamics vs. Static Dimensions):** Traditional dual attention (like DANet/DaViT) typically focuses on capturing dependencies across different **dimensions** (e.g., spatial vs. channel) within a single feature map. In contrast, our mechanism focuses on the **temporal stages** of the diffusion process.
> * **Goal (Structure/Semantic Disentanglement):** Our block introduces two parallel cross-attention branches that attend to distinct **diffusion timesteps** of the source asset: one captures fine-grained structural features ($t \approx 0$) for identity preservation, while the other captures semantic transition features ($t \approx 1$) to guide the intended edit.
> * **Mechanism (Time-Adaptive Gating):** Unlike static dimensional attention, we employ a TAG mechanism to dynamically balance these two guidance signals across denoising steps. This ensures semantic alignment in early stages and structural fidelity in later stages, which is a unique requirement for consistent 3D editing.
>
> We will revise the manuscript to explicitly detail these architectural and conceptual distinctions, underscoring the novelty of our contribution.

---

> > ### Author Rebuttal · Reviewer_bcYR · 2026-04-06
> >
> > Thanks to the authors for the detailed rebuttal. It resolves my main concerns.

---

> > > ### Author Response · Authors · 2026-04-07
> > >
> > > Thank you for you review. We are pleased that our detailed response has resolved your key concerns.

---

### Official Review · Reviewer_56WL · 2026-03-12

**Soundness:** 3
**Presentation:** 2
**Significance:** 4
**Originality:** 4
**Overall Recommendation:** 5
**Confidence:** 4

**Summary:**

The paper proposes a new dataset and method for 3D editing task. Their dataset, 3DEditVerse, includes thousands of i̇nput-output pairs that are lifted from image domain (generated using Flux.2) to 3D domain (using Trellis). In order to make sure that the edited regions are structure-preserving they apply automated localization & segmentation and repaint approach using TRELLIS. They additionally propose a method, 3DEditFormer, for 3D editing leveraging their large-scale dataset. The method includes a dual attention block and time-adaptive gating in order to achieve localized editing.

**Compliance With Llm Reviewing Policy:**

Affirmed.

**Final Justification:**

Rebuttal addresses my concerns. I raised my score to Accept.

**Key Questions For Authors:**

* Are the dark/shadowed regions of the shapes because of the lighting conditions of the rendering, which can be easily fixed by more ambient lighting. Or is it because the images generated create those artifacts when lifted to 3D? That is important because the shapes would ideally have minimal baked-in lighting to be useful.
* Did you consider using additional metrics such as directional CLIP score and VLM evaluation? Since the current metrics assume there is a certain ground truth for an editing task. However, since this is a text-based editing the generated result can be different from ground truth but also reflect the editing prompt well. That's why I believe CLIP-based or VLM-based metrics would be useful.
* Is the method compared against the default TRELLIS editing without their components? If one of the entries in the tables correspond to that, it would be useful to clarify that.

**Limitations:**

yes

**Strengths And Weaknesses:**

Strengths:
* Soundness & Significance:
    * Overall, dataset contribution is extremely useful since many methods use inference-only approaches such as prompt-to-prompt to tackle the editing problem, without requiring large-scale dataset for training. The dataset would give more freedom on developing better methods by learning from such distributions. That's a very useful contribution. The way the data is collected seems sound and correct.
    * Additionally, they propose an editing method, for which they propose additional blocks for TRELLIS method, it's also a valid approach.
* Presentation
    * Writing is easy-to-follow and there's good transition from dataset part to method part .
* Originality:
    * Most important part to me is providing a new dataset for editing and their data collection method makes use of image diffusion models with which they can generate an abundance of data.

Weaknesses
* Soundness:
    * I have some doubts about the choices of the metrics. Even though there is a ground truth i̇nput-output pair for each editing task, I don’t think chamfer distance, psnr, ssim, lpips make a lot of sense; because when the prompt is “wearing a christmas hat”, the hat that is generated can be extremely different from the ground truth one. That being sad, I think they’re useful to test if the most of the i̇nput shape is preserved, however, a method which always outputs the i̇nput may be advantaged in those metrics. That’s why it would be useful to have some additional metrics like directional CLIP score or VLM-based evaluation.
     * The proposed method is an improvement on top of TRELLIS, but I'm not sure if the default TRELLIS editing is compared in the paper. Does the baseline in ablation study correspond to that?
* Presentation
    * The renderings of the 3D shapes are a bit difficult to follow and see as they have strong shadowing hiding many features of the shape. That is, the overall dark appearance of the shapes make it difficult to identify.
    * Limited discussion on the main results, in section 5, it could provide more in-depth analysis of the results.

---

> ### Author Rebuttal · Authors · 2026-03-30
>
> **To Reviewer 56WL**
>
> We appreciate your insightful and constructive feedback. In the following, we provide our point-by-point response and hope our response helps address your concerns. We also look forward to the subsequent discussion which further helps to solve the current issues.
>
> > **Q1: Evaluation with directional CLIP score and VLM.**
>
> **A1:**  We appreciate the reviewer's insightful suggestion and agree that semantic alignment is a crucial evaluation axis for text-based editing, complementing our existing structural metrics. To address this, we have now evaluated all methods using both **Directional CLIP Score** and a **VLM-based score**:
>
> * **Directional CLIP Score:** Evaluates whether the change in the 3D asset aligns with the change specified in the text prompt.
> * **VLM-based Score:** Utilizes a Vision-Language Model (Qwen3-VL-Plus) to provide a holistic score on how well the final 3D result adheres to the editing instructions.
>
> | Method | Directional CLIP ↑ | VLM Score ↑ |
> | :--- | :---: | :---: |
> | EditP23 | 0.056 | 3.09 |
> | Instant3dit | 0.075 | 3.92 |
> | VoxHammer | 0.107 | 6.01 |
> | **3DEditFormer (Ours)** | **0.113** | **6.16** |
>
> As the results show, 3DEditFormer outperforms all baselines in semantic alignment, reinforcing its superior balance of structural integrity and fidelity to the user's edit instruction. We will add these results and analysis to the revised manuscript.
>
> > **Q2: Visual presentation of shadowed/dark rendering artifacts.**
>
> **A2:** We clarify that the darkened regions are not artifacts inherent to any specific 3D editing method, but rather a natural consequence of the Cycles path-tracing engine used during rendering.
>
> To demonstrate this, we provide a "Comparison between lit and unlit rendering modes" section in https://anonymousresearch37.github.io/3DEditFormer/. By switching to an unlit/flat shading mode (which displays raw colors without light interaction), it is clear that:
>
> * The darkened areas disappear, revealing clean and consistent textures.
> * The geometry is free of "baked-in" shadows or reconstruction artifacts.
>
> This confirms the darkened areas are rendering-induced shadows, not flaws in our model's output or any other editing method's output.
>
> > **Q3: Comparison with default TRELLIS editing.**
>
> **A3:** There is a fundamental difference between default TRELLIS editing and our ablation baseline:
>
> * **Default TRELLIS Editing:** This is a training-free approach that relies on the Repaint algorithm and explicit 3D masks to constrain modifications.
> * **Ablation Baseline:** The "Baseline" in Table 3 refers to a version of the Trellis architecture that has been supervisedly trained on our proposed 3DEdit Verse dataset, but without our specific Dual-Guidance Attention and Time-Adaptive Gating components.
>
> To demonstrate the superiority of our learned approach, we have evaluated our **3DEditFormer** against the default **Trellis Repainting** method. Our trained, mask-free approach consistently outperforms the original training-free, mask-dependent pipeline across all metrics:
>
> | Method | CD ↓ | NC ↑ | $F1^{0.01}$ ↑ | PSNR ↑ | SSIM ↑ | LPIPS ↓ | DINO-I ↑ |
> | :--- | :---: | :---: | :---: | :---: | :---: | :---: | :---: |
> | Trellis Repainting | 10.92 | 0.871 | 74.22 | 25.77 | 0.936 | 0.055 | 0.956 |
> | 3DEditFormer (Ours) | **7.04** | **0.904** | **86.05** | **26.42** | **0.938** | **0.045** | **0.962** |
>
> These results powerfully demonstrate that our model not only simplifies the user workflow by eliminating the need for 3D masks but also achieves substantially higher geometric and textural fidelity.
>
> > **Q4: Limited in-depth analysis of the main results.**
>
> **A4:** We agree and thank the reviewer for this valuable feedback. Due to strict page constraints in the initial submission, the discussion of our experimental findings was kept relatively concise. In the revised manuscript, we will dedicate more space in Section 5 to provide:
>
> * A more granular discussion of the quantitative results, linking specific metric improvements to our architectural choices.
> * A qualitative analysis highlighting specific failure cases of SoTA methods where our 3DEditFormer succeeds, further illustrating the practical advantages of our approach.

---

> > ### Author Rebuttal · Reviewer_56WL · 2026-04-02
> >
> > Thank you very much. The rebuttal addresses my concerns, I'll raise my score to Accept.

---

> > > ### Author Response · Authors · 2026-04-02
> > >
> > > Thank you for your supportive response and for accepting our paper. We highly value your time and constructive suggestions during the review.

---

### Official Review · Reviewer_9SPP · 2026-03-15

**Soundness:** 3
**Presentation:** 4
**Significance:** 3
**Originality:** 3
**Overall Recommendation:** 5
**Confidence:** 4

**Summary:**

This paper proposes a two‑part contribution to the field of 3D editing. First, it introduces 3DEditVerse, a large paired dataset of 3D assets generated automatically using pose‑driven geometry edits and text‑guided appearance edits. Second, it presents 3DEditFormer, a transformer‑based editing model built on top of a frozen Trellis backbone. 3DEditFormer injects structural cues and early‑stage semantic transition cues through a Dual‑Guidance Attention block, with a Time‑Adaptive Gating mechanism that balances these signals across diffusion steps. Contrary to many existing solutions, the approach is mask‑free at inference time.

**Compliance With Llm Reviewing Policy:**

Affirmed.

**Final Justification:**

The paper and rebuttal are solid, and I recommend acceptance

**Key Questions For Authors:**

1. How is the Repaint strategy integrated with Trellis during denoising, and does it interact with the DualAttn module?
2. The pipeline uses multiple models, all of which can introduce incorrect elements. How do you deal with artifacts in generated masks/3D assets during dataset generation process?
3. Have you evaluated the model on real‑world scanned assets or on images with different lighting conditions?
4. What GPU requirements does the proposed method require (i.e., VRAM)?
5. Figure 3 suggests that the edit is applied to only a single image, as there is only one I_tgt shown. This raises the question of whether the specific view to which the edit is applied affects the outcome. For example, if the edit is performed on a view from behind the rabbit while attempting to add a bow tie, the 3D object edit might fail or produce an unnatural result.

**Limitations:**

yes

**Strengths And Weaknesses:**

Strengths:
1. 3DEditVerse provides a large, paired 3D editing benchmark.
2. Dual‑Guidance Attention removes the need for expensive 3D masks.
3. The dual‑stage feature extraction and adaptive gating are simple, well‑motivated, and effective.
4. Multiple 3D and 2D metrics, baseline comparisons, mask‑sensitivity analysis, and ablations.
5. The paper presents a clear and well-structured pipeline for single-image 3D avatar generation.

Weaknesses:
1. Heavy reliance on multiple foundation models (Flux, Qwen‑VL, SAM2, Trellis) during dataset creation process.
2. No tests on scanned or real‑world assets, generalization beyond synthetic edits is not shown. Even in Appendix H Visualization on Real-Object Editing, the objects look synthetic.
3. No inference speed or memory consumption reported for the proposed 3DEditFormer model.
4. High‑frequency geometric detail preservation is not explicitly measured.
5. It seems that the filtering steps illustrated in Figure 2 and the Trellis Repaint are not described in sufficient detail in the main paper, which reduces the overall clarity.

---

> ### Author Rebuttal · Authors · 2026-03-30
>
> **To Reviewer 9SPP**
>
> We appreciate your insightful and constructive feedback. In the following, we provide our point-by-point response and hope our response helps address your concerns. We also look forward to the subsequent discussion which further helps to solve the current issues.
>
> > **Q1: Generalization beyond synthetic edits.**
>
> **A1:** We conducted additional 3D editing experiments on the OmniObject3D dataset (CVPR23--*OmniObject3D: Large-Vocabulary 3D Object Dataset for Realistic Perception, Reconstruction and Generation*), which comprises thousands of high-quality real-world scanned objects. As presented on the "3DEditFormer | Real-World Scanned Assets Editing" section at https://anonymousresearch37.github.io/3DEditFormer/, our method consistently produces precise and consistent edits on these complex assets, demonstrating robust generalization beyond synthetic data.
>
> > **Q2: Inference speed and memory consumption.**
>
> **A2:** As summarized in the table below, 3DEditFormer is significantly faster than SoTA VoxHammer (~20s vs. ~65s) while maintaining competitive memory usage. All experiments were conducted on a single NVIDIA L40s GPU.
>
> | Method | Runtime (s) | VRAM (GB) |
> | - | - | - |
> | EditP23 | 9.8 | 28.35 |
> | Instant3Dit | 16.7 | 41.47 |
> | VoxHammer | 64.8 | 12.86 |
> | 3DEditFormer (Ours) | 19.8 | 30.27 |
>
> > **Q3: High-frequency geometric detail preservation.**
>
> **A3:** To quantitatively address this, we introduced two additional metrics specifically targeting local surface variations and high-frequency information:
>
> * **Laplacian Error**: This captures the difference in differential coordinates ($L(x_i) = x_i - \frac{1}{|N_i|} \sum_{j \in N_i} x_j$) between predicted and ground-truth meshes. It directly reflects local surface bending and is highly sensitive to unintended smoothing.
> * **Curvature Error**: We measure the Root Mean Square Error (RMSE) of Mean Curvature. This identifies whether sharp edges or intricate features are lost, as these correspond to high-curvature regions.
>
> As shown below, **3DEditFormer** substantially outperforms all baselines on both metrics, confirming its superior ability to preserve geometric integrity:
>
> | Method | Laplacian Error (1e-3) ↓ | Curvature Error (1e-3) ↓ |
> | :--- | :---: | :---: |
> | EditP23 | 14.94 | 5.62 |
> | Instant3dit | 18.30 | 4.88 |
> | VoxHammer | 10.19 | 4.31 |
> | **3DEditFormer (Ours)** | **8.82** | **4.09** |
>
> > **Q4: Insufficient details for filtering steps and Trellis Repaint.**
>
> **A4:** We agree that these details are crucial. We will revise the manuscript to move the following technical descriptions from the Appendix to the main body, ensuring a self-contained and comprehensive explanation:
>
> * **Consistency Filtering:** We will elaborate on the multi-view voting strategy (70 views) and the DINOv2 similarity-based threshold used to automatically prune artifacts and identity mismatches.
> * **Trellis Repaint:** We will incorporate the formal latent-space fusion equation, $\hat{z}\_{t} = M\_{3D} \odot z\_{t}^{tgt}+(1 - M\_{3D}) \odot z\_{t}^{src}$, explaining how the 3D mask $M_{3D}$ anchors unedited regions to the source latent.
>
> > **Q5: Integration of Repaint with Trellis and its interaction with the DualAttn module..**
>
> **A5:** We would like to clarify that the **Repaint strategy** and the **DualAttn module** serve distinct, non-interacting roles at different stages:
>
> * **Repaint Strategy (Dataset Creation):** Repaint is an algorithm used exclusively within our **data generation pipeline** to construct the 3DEdit Verse dataset. It ensures that the generated ground-truth pairs are spatially confined and consistent by fusing source and target latents within a 3D mask during the initial lifting process.
> * **DualAttn Module (Model Architecture):** In contrast, DualAttn is a novel component of our proposed **3DEditFormer**. It is designed to inject structural priors from a source asset into the target generation process via cross-attention pathways during inference.
>
> In short, Repaint is a pre-processing tool for data creation, whereas DualAttn is an integral architectural component for inference.
>
> > **Q6: Dependence on the specific viewpoint of $I^{tgt}$.**
>
> **A6:** To investigate this, we conducted experiments across multiple viewpoints (front, side, and back), which are presented in the "3DEditFormer | Different View Editing" section on https://anonymousresearch37.github.io/3DEditFormer/.
>
> Our results confirm that, for a successful edit, the target region must be reasonably visible in the conditioning image $I^{tgt}$. For example, "adding a bell to a cow's neck" fails from a rear view where the neck is occluded. Importantly, we found this is a common limitation shared by SoTA methods like VoxHammer. Therefore, a practical requirement of these frameworks is that the user provides a 2D edit on an informative view.
>
> > **Q7: Error propagation during dataset creation.**
>
> **A7:** Please refer to **A1** provided in the rebuttal to **Reviewer bcYR**.

---

> > ### Author Rebuttal · Reviewer_9SPP · 2026-04-02
> >
> > Thanks to the authors for the detailed rebuttal. It resolves my main concerns.  I will raise my score to Accept.

---

> > > ### Author Response · Authors · 2026-04-02
> > >
> > > Thank you very much for your positive feedback and raising your score to Accept. We truly appreciate your time and valuable comments throughout the review process.

---

### Decision · Program_Chairs · 2026-04-30

**Decision:**

Accept (regular)

**Comment:**

The paper introduces 3DEditVerse, a large-scale paired 3D editing benchmark, and 3DEditFormer, a mask-free transformer for 3D editing built on Trellis with dual-guidance attention and time-adaptive gating. All reviewers recognize the dataset as a valuable contribution to the community and agree that the overall pipeline is well-motivated. The strong quantitative results suggest state-of-the-art performance over existing baselines. Initially, the reviewers raised concerns about evaluation metrics, comparison fairness, novelty of the dual attention design, and real-world generalization. The authors seem to have sufficiently addressed these points with additional experiments and clarifications. All three reviewers confirmed their concerns were fully resolved, with final ratings of 5, 5, 4. The authors committed to expanding the main text with additional details, adding in-depth result analysis in Section 5, and clarifying the novelty of their dual attention design.